# PROPER LAPLACIAN REPRESENTATION LEARNING

**Diego Gomez, Michael Bowling**[*]**, Marlos C. Machado**[*]
Department of Computing Science, University of Alberta
Alberta Machine Intelligence Institute (Amii)
[*] Canada CIFAR AI Chair
Edmonton, AB T6G 2R3, Canada
{gomeznor,mbowling,machado}@ualberta.ca

## ABSTRACT

The ability to learn good representations of states is essential for solving large reinforcement learning problems, where exploration, generalization, and transfer are particularly challenging. The *Laplacian representation* is a promising approach to address these problems by inducing informative state encoding and intrinsic rewards for temporally-extended action discovery and reward shaping. To obtain the Laplacian representation one needs to compute the eigensystem of the graph Laplacian, which is often approximated through optimization objectives compatible with deep learning approaches. These approximations, however, depend on hyperparameters that are impossible to tune efficiently, converge to arbitrary rotations of the desired eigenvectors, and are unable to accurately recover the corresponding eigenvalues. In this paper we introduce a theoretically sound objective and corresponding optimization algorithm for approximating the Laplacian representation. Our approach naturally recovers both the true eigenvectors and eigenvalues while eliminating the hyperparameter dependence of previous approximations. We provide theoretical guarantees for our method and we show that those results translate empirically into robust learning across multiple environments.

## 1 INTRODUCTION

Reinforcement learning (RL) is a framework for decision-making where an agent continually takes actions in its environment and, in doing so, controls its future states. After each action, given the current state and the action itself, the agent receives a reward and a next state from the environment. The objective of the agent is to maximize the sum of these rewards. In principle, the agent has to visit all states and try all possible actions a reasonable number of times to determine the optimal behavior. However, in complex environments, e.g., when the number of states is large or the environment changes with time, this is not a plausible strategy. Instead, the agent needs the ability to learn representations of the state that facilitate exploration, generalization, and transfer.

The *Laplacian framework* (Mahadevan, 2005; Mahadevan & Maggioni, 2007) proposes one such representation. This representation is based on the graph Laplacian, which, in the tabular case, is a matrix that encodes the topology of the state space based on both the policy the agent uses to select actions and the environment dynamics. Specifically, the $d-$dimensional *Laplacian representation* is a map from states to vectors whose entries correspond to $d$ eigenvectors of the Laplacian.

Among other properties, the Laplacian representation induces a metric space where the Euclidean distance of two representations correlates to the temporal distance of their corresponding states; moreover, its entries correspond to directions that maximally preserve state value information (Petrik, 2007). Correspondingly, it has been used for reward shaping (Wu et al., 2019; Wang et al., 2023), as a state representation for value approximation (e.g., Mahadevan & Maggioni, 2007; Lan et al., 2022; Wang et al., 2022; Farebrother et al., 2023), as a set of intrinsic rewards for exploration via temporally-extended actions (see overview by Machado et al., 2023), zero-shot learning (Touati et al., 2023), and to achieve state-of-the-art performance in sparse reward environments (Klissarov & Machado, 2023).

When the number of states, $|\mathcal{S}|$, is small, the graph Laplacian can be represented as a matrix and one can use standard matrix eigendecomposition techniques to obtain its *eigensystem* and the cor-

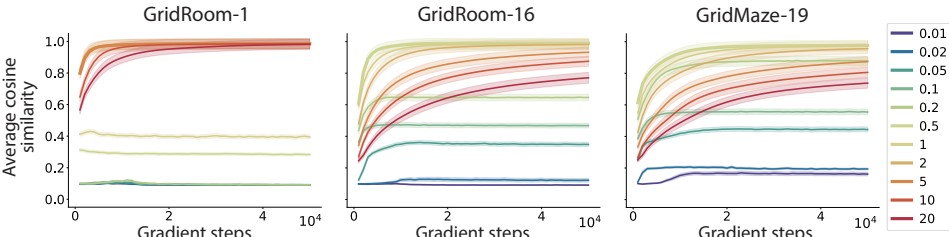

Figure 1: Average cosine similarity between the true Laplacian representation and GGDO for different values of the barrier penalty coefficient, averaged over 60 seeds, with the best coefficient highlighted. The shaded region corresponds to a 95% confidence interval.

responding Laplacian representation. In practice, however, $|\mathcal{S}|$ is large, or even uncountable. Thus, at some point it becomes infeasible to directly compute the eigenvectors of the Laplacian. In this context, Wu et al. (2019) proposed a scalable optimization procedure to obtain the Laplacian representation in state spaces with uncountably many states. Such an approach is based on a general definition of the graph Laplacian as a linear operator, also introduced by Wu et al. (2019). Importantly, this definition allows us to model the Laplacian representation as a neural network and to learn it by minimizing an unconstrained optimization objective, the *graph drawing objective* (GDO).

A shortcoming of GDO, however, is that arbitrary rotations of the eigenvectors of the Laplacian minimize the graph drawing objective (Wang et al., 2021). This is, in general, undesirable since rotating the eigenvectors affect the properties that make them useful as intrinsic rewards (see Appendix C for more details). As a solution, Wang et al. (2021) proposed the *generalized graph drawing objective* (GGDO), which breaks the symmetry of the optimization problem by introducing a sequence of decreasing hyperparameters to GDO. The true eigenvectors are the only solution to this new objective. Despite this, when minimizing this objective with stochastic gradient descent, the rotations of the smallest eigenvectors[1] are still equilibrium points of the generalized objective. Consequently, there is variability in the eigenvectors one actually finds when minimizing such an objective, depending, for example, on the initialization of the network and on the hyperparameters chosen.

These issues are particularly problematic because it is **impossible to tune the hyperparameters** of GGDO without already having access to the problem solution: previous results, when sweeping hyperparameters, used the cosine similarity between the *true eigenvectors* and the approximated solution as a performance metric. To make matters worse, the best **hyperparameters are environment dependent**, as shown in Figure 1. Thus, when relying on GDO, or GGDO, *it is impossible to guarantee an accurate estimate of the eigenvectors of the Laplacian in environments where one does not know these eigenvectors in advance*, which obviously defeats the whole purpose. Finally, the existing objectives are **unable to approximate the eigenvalues** of the Laplacian, and existing heuristics heavily depend on the accuracy of the estimated eigenvectors (Wang et al., 2023).

In this work, we introduce a theoretically sound max-min objective and a corresponding optimization procedure for approximating the Laplacian representation that addresses *all* the aforementioned issues. Our approach naturally recovers both the true eigenvectors and eigenvalues while eliminating the hyperparameter dependence of previous approximations. Our objective, which we call the Augmented Lagrangian Laplacian Objective (ALLO), corresponds to a Lagrangian version of GDO augmented with stop-gradient operators. These operators break the symmetry between the rotations of the Laplacian eigenvectors, turning the eigenvectors and eigenvalues into the unique stable equilibrium point of min-max ALLO under gradient ascent-descent dynamics, independently of the original hyperparameters of GGDO. Besides theoretical guarantees, we empirically demonstrate that our proposed approach is robust across different environments with different topologies and that it is able to accurately recover the eigenvalues of the graph Laplacian as well.

## 2 BACKGROUND

We first review the reinforcement learning setting, before presenting the Laplacian representation and previous work on optimization objectives for its approximating approximation.

---

[1]We refer to the eigenvectors with corresponding smallest eigenvalues as the "smallest eigenvectors".

**Reinforcement Learning.** We consider the setting in which an agent interacts with an environment. The environment is a reward-agnostic Markov-decision process $M = (\mathcal{S}, \mathcal{A}, P, \mu_0)$ with finite state space $\mathcal{S} = \{1, \cdots, |\mathcal{S}|\}$,[2] finite action space $\mathcal{A} = \{1, \cdots, |\mathcal{A}|\}$, transition probability map $P : \mathcal{S} \times \mathcal{A} \to \Delta(\mathcal{S})$, which maps a state-action pair $(s, a)$ to a state distribution $P(\cdot|s, a)$ in the simplex $\Delta(\mathcal{S})$, and initial state distribution $\mu_0 \in \Delta(\mathcal{S})$. The agent is characterized by the policy $\pi : \mathcal{S} \to \Delta(\mathcal{A})$ that it uses to choose actions. At time-step $t = 0$, an initial state $S_0$ is sampled from $\mu_0$. Then, the agent samples an action $A_0$ from its policy and, as a response, the environment transitions to a new state $S_1$, following the distribution $P(S_0, A_0)$. After this, the agent selects a new action, the environment transitions again, and so on. The agent-environment interaction determines a Markov process characterized by the transition matrix $\mathbf{P}_\pi$, where $(\mathbf{P}_\pi)_{s, s'} = \sum_{a \in \mathcal{A}} \pi(s, a) P(s'|s, a)$ is the probability of transitioning from state $s$ to state $s'$ while following policy $\pi$.

**Laplacian Representation.** In graph theory, the object of study is a node set $\mathcal{V}$ whose elements are pairwise connected by edges. The edge between a pair of nodes $v, v' \in \mathcal{V}$ is quantified by a non-negative real number $w_{v, v'}$, which is 0 only if there is no edge between the nodes. The adjacency matrix, $\mathbf{W} \in \mathbb{R}^{|\mathcal{V}| \times |\mathcal{V}|}$, stores the information of all edges such that $(\mathbf{W})_{v, v'} = w_{v, v'}$. The degree of a node $v$ is the sum of the adjacency weights between $v$ and all other nodes in $\mathcal{V}$ and the degree matrix $\mathbf{D} \in \mathbb{R}^{|\mathcal{V}| \times |\mathcal{V}|}$ is the diagonal matrix containing these degrees. The Laplacian $\mathbf{L}$ is defined as $\mathbf{L} = \mathbf{D} - \mathbf{W}$, and, just as the adjacency matrix, it fully encodes the information of the graph.

If we consider the state space of an MDP $M$ as the set of nodes, $\mathcal{V} = \mathcal{S}$, and $\mathbf{W}$ as determined by $\mathbf{P}_\pi$, then we might expect the graph Laplacian to encode useful temporal information about $M$, meaning the number of time steps required to go from one state to another. In accordance with Wu et al. (2019), we broadly define the *Laplacian* in the tabular reinforcement learning setting as any matrix $\mathbf{L} = \mathbf{I} - f(\mathbf{P}_\pi)$, where $f : \mathbb{R}^{|\mathcal{S}| \times |\mathcal{S}|} \to \mathrm{Sym}_{|\mathcal{S}|}(\mathbb{R})$ is some function that maps $\mathbf{P}_\pi$ to a symmetric matrix.[3] For example, if $\mathbf{P}_\pi$ is symmetric, the Laplacian is typically defined as either $\mathbf{L} = \mathbf{I} - \mathbf{P}_\pi$ or $\mathbf{L} = \mathbf{I} - (1 - \lambda)\mathbf{\Phi}_\pi^\lambda$, where $\mathbf{\Phi}_\pi^\lambda = (\mathbf{I} - \lambda\mathbf{P}_\pi)^{-1}$ is a matrix referred to as the successor representation matrix (Dayan, 1993; Machado et al., 2018). In the case where $\mathbf{P}_\pi$ is not symmetric, $\mathbf{L}$ is usually defined as $\mathbf{L} = \mathbf{I} - \frac{1}{2}(\mathbf{P}_\pi + \mathbf{P}_\pi^\top)$ to ensure it is symmetric (Wu et al., 2019).

The *Laplacian representation*, $\phi : \mathcal{S} \to \mathbb{R}^d$, maps a state $s$ to $d$ corresponding entries in the set of $0 < d \leq |\mathcal{S}|$ smallest eigenvectors of $\mathbf{L}$, i.e., $\phi(s) = [\mathbf{e}_1[s], \cdots, \mathbf{e}_d[s]]^\top$, where $\mathbf{e}_i$ is the $i-$th smallest eigenvector of $\mathbf{L}$ and $\mathbf{e}_i[s]$, its $s-$th entry (Mahadevan & Maggioni, 2007; Stachenfeld et al., 2014; Machado et al., 2017).

**The Graph Drawing Objective.** Given the graph Laplacian $\mathbf{L}$, the spectral graph drawing optimization problem (Koren, 2003) is defined as follows:

$$\min_{\mathbf{u}_1, \cdots, \mathbf{u}_d \in \mathbb{R}^\mathcal{S}} \quad \sum_{i=1}^d \langle \mathbf{u}_i, \mathbf{L}\mathbf{u}_i \rangle \tag{1}$$

$$\text{such that} \quad \langle \mathbf{u}_j, \mathbf{u}_k \rangle = \delta_{jk} \, , \; 1 \leq k \leq j \leq d \, ,$$

where $\langle \cdot, \cdot \rangle$ is the inner product in $\mathbb{R}^{|\mathcal{S}|}$ and $\delta_{jk}$ is the Kronecker delta. This optimization problem has two desirable properties. The first one is that the $d$ smallest eigenvectors of $\mathbf{L}$ are a global optimizer.[4] Hence, the Laplacian representation $\phi$ associated with $\mathbf{L}$ is a solution to this problem. The second property is that both objective and constraints can be expressed as expectations, making the problem amenable to stochastic gradient descent. In particular, the original *constrained* optimization problem (1) can be approximated by the unconstrained *graph drawing objective* (GDO):

$$\min_{\mathbf{u} \in \mathbb{R}^{d|\mathcal{S}|}} \quad \sum_{i=1}^d \langle \mathbf{u}_i, \mathbf{L}\mathbf{u}_i \rangle + b \sum_{j=1}^d \sum_{k=1}^d \left( \langle \mathbf{u}_j, \mathbf{u}_k \rangle - \delta_{jk} \right)^2 \, , \tag{2}$$

---

[2]For ease of exposition, we restrict the notation, theorems, and proofs to the finite state space setting. However, we generalize all of them to the abstract setting in Appendix A.5, where the state space is an abstract measure space, in a similar fashion as in the work by Wu et al. (2019) and Wang et al. (2021).

[3]The Laplacian has $|S|$ different **real** eigenvectors and corresponding eigenvalues only if it is symmetric.

[4]For proofs in the tabular setting, see the work by Koren (2003) for the case $d = 2$, and Lemma 1 for arbitrary $d$. For the abstract setting, see the work by Wang et al. (2021).

where $b \in (0, \infty)$ is a scalar hyperparameter and $\mathbf{u} = [\mathbf{u}_1^\top, \cdots, \mathbf{u}_d^\top]^\top$ is the vector that results from concatenating the vectors $(\mathbf{u}_i)_{i=1}^d$ (Wu et al., 2019).

**The Generalized Graph Drawing Objective.** As mentioned before, any rotation of the smallest eigenvectors of the Laplacian $\mathbf{L}$ is a global optimizer of the constrained optimization problem (1). Hence, even with an appropriate choice of hyperparameter $b$, GDO does not necessarily approximate the Laplacian representation $\phi$. As a solution, Wang et al. (2021) present the generalized graph drawing optimization problem:

$$\min_{\mathbf{u} \in \mathbb{R}^{d|\mathcal{S}|}} \quad \sum_{i=1}^d c_i \langle \mathbf{u}_i, \mathbf{L}\mathbf{u}_i \rangle \tag{3}$$
$$\text{such that} \quad \langle \mathbf{u}_j, \mathbf{u}_k \rangle = \delta_{jk} , \ 1 \le k \le j \le d,$$

where $c_1 > \cdots > c_d > 0$ is a monotonically decreasing sequence of $d$ hyperparameters. Correspondingly, the unconstrained *generalized graph drawing objective* (GGDO) is defined as:

$$\min_{\mathbf{u} \in \mathbb{R}^{d|\mathcal{S}|}} \quad \sum_{i=1}^d c_i \langle \mathbf{u}_i, \mathbf{L}\mathbf{u}_i \rangle + b \sum_{j=1}^d \sum_{k=1}^d \min(c_j, c_k) \big( \langle \mathbf{u}_j, \mathbf{u}_k \rangle - \delta_{jk} \big)^2 . \tag{4}$$

Wang et al. (2021) prove that the optimization problem (3) has a unique global minimum that corresponds to the smallest eigenvectors of $\mathbf{L}$, for *any* possible choice of the hyperparameter sequence $(c_i)_{i=1}^d$. However, in the unconstrained setting, which is the setting used when training neural networks, these hyperparameters do affect both the dynamics and the quality of the final solution. In particular, Wang et al. (2021) found in their experiments that the linearly decreasing choice $c_i = d - i + 1$ performed best across different environments. More importantly, under gradient descent dynamics, the introduced coefficients are unable to break the symmetry and arbitrary rotations of the eigenvectors are still equilibrium points (see Corollary (1) in Section 4).

**The Abstract and Approximate Settings.** Lastly, the previous optimization problems (1)-(4) can be generalized to the abstract setting, where $\mathcal{S}$ is potentially an uncountable measure space (e.g., the continuous space $\mathbb{R}^n$). In practice, the only thing that changes is that finite-dimensional vectors $\mathbf{u}_i$ are replaced by real-valued functions $u_i : \mathcal{S} \to \mathbb{R}$,[5] and matrices by linear operators that map these functions to functions in the same space (see Appendix A.5 for a detailed discussion). This generalization is useful because it allows to introduce function approximators in a principled way. Specifically, we replace the $d$ functions $(u_1, \cdots, u_d)$ by the parametric model $\phi_{\boldsymbol{\theta}} : \mathcal{S} \to \mathbb{R}^d$, where $\boldsymbol{\theta}$ is a finite dimensional parameter vector. In our case, $\phi_{\boldsymbol{\theta}}$ represents a neural network, and $\boldsymbol{\theta}$ a vector containing the weights of the network. This choice allows us to find an approximate solution by iteratively sampling a transition batch, calculating the corresponding optimization objective, and propagating the gradients by means of any stochastic gradient descent-based optimizer. In the following chapters we will see why this gradient descent procedure is incompatible with GGDO and how our proposed objective, ALLO, overcomes this incompatibility in theory and in practice.

## 3 Augmented Lagrangian Laplacian Objective

In this section we introduce a method that retains the benefits of GGDO while avoiding its pitfalls. Specifically, we relax the goal of having a unique global minimum for a constrained optimization problem like (3). Instead, we modify the stability properties of the unconstrained dynamics to ensure that the only stable equilibrium point corresponds to the Laplacian representation.

**Asymmetric Constraints as a Generalized Graph Drawing Alternative.** We want to break the *dynamical symmetry* of the Laplacian eigenvectors that make any of their rotations an equilibrium point for GDO (2) and GGDO (4) while avoiding the use of hyperparameters. For this, let us consider the original graph drawing optimization problem (1). If we set $d = 1$, meaning we try to approximate only the first eigenvector $\mathbf{e}_1$, it is clear that the only possible solution is $\mathbf{u}_1^* = \mathbf{e}_1$. This happens because the only possible rotations are $\pm \mathbf{e}_1$. If we then try to solve the optimization problem for

---

[5]As a consequence, the eigenvectors of the Laplacian are commonly called eigenfunctions. However, technically speaking, the eigenfunctions are still eigenvectors in the space of square integrable functions.

$d = 2$, but fix $\mathbf{u}_1 = \mathbf{e}_1$, the solution will be $(\mathbf{u}_1^*, \mathbf{u}_2^*) = (\mathbf{e}_1, \mathbf{e}_2)$, as desired. Repeating this process $d$ times, we can obtain $\phi$. Thus, we can eliminate the need for the $d$ hyperparameters introduced by GGDO by solving $d$ separate optimization problems. To replicate this separation while maintaining a single unconstrained optimization objective, we introduce the stop-gradient operator $[\![\cdot]\!]$ in GDO. This operator does not affect the objective, but it indicates that, when following gradient dynamics, the real gradient of the objective is not used. Instead, when calculating derivatives, whatever is inside the operator is assumed to be constant. Specifically, the objective becomes:

$$\min_{\mathbf{u} \in \mathbb{R}^{d|\mathcal{S}|}} \quad \sum_{i=1}^{d} \langle \mathbf{u}_i, \mathbf{L}\mathbf{u}_i \rangle + b \sum_{j=1}^{d} \sum_{k=1}^{j} \left( \langle \mathbf{u}_j, [\![\mathbf{u}_k]\!] \rangle - \delta_{jk} \right)^2. \tag{5}$$

Note that in addition to the stop-gradient operators, the upper limit in the inner summation is now the variable $j$, instead of the constant $d$. These two modifications ensure that $\mathbf{u}_i$ changes only to satisfy the constraints associated to the previous vectors $(\mathbf{u}_j)_{j=1}^{i-1}$ and itself, but not the following ones, i.e., $(\mathbf{u}_j)_{j=i+1}^{d}$. Hence, the asymmetry in the descent direction achieves the same effect as having $d$ separate optimization problems. In particular, as proved in Lemma 2 in the next section, the descent direction of the final objective, yet to be defined, becomes $\mathbf{0}$ only for permutations of a subset of the Laplacian eigenvectors, and not for any of its rotations.

**Augmented Lagrangian Dynamics for Exact Learning.** The regularization term added in all of the previous objectives (2), (4), and (5) is typically referred to as a quadratic penalty with barrier coefficient $b$. This coefficient shifts the equilibrium point of the original optimization problems (1) and (3), and one can only guarantee that the desired solution is obtained in the limit $b \to \infty$ (see Chapter 17 by Nocedal & Wright, 2006). In practice, one can increase $b$ until a satisfactory solution is found. However, not only is there no direct metric to tell how close one is to the true solution, but also an extremely large $b$ is empirically bad for neural networks when optimizing GDO or GGDO. As a principled alternative, we propose the use of augmented Lagrangian methods. Specifically, we augment the objective (5) by adding the original constraints, multiplied by their corresponding dual variables, $(\beta_{jk})_{1 \le k \le j \le d}$. This turns the optimization problem into the following max-min objective, which we call the *augmented Lagrangian Laplacian objective* (ALLO):

$$\max_{\boldsymbol{\beta}} \min_{\mathbf{u} \in \mathbb{R}^{d|\mathcal{S}|}} \sum_{i=1}^{d} \langle \mathbf{u}_i, \mathbf{L}\mathbf{u}_i \rangle + \sum_{j=1}^{d} \sum_{k=1}^{j} \beta_{jk} \left( \langle \mathbf{u}_j, [\![\mathbf{u}_k]\!] \rangle - \delta_{jk} \right) + b \sum_{j=1}^{d} \sum_{k=1}^{j} \left( \langle \mathbf{u}_j, [\![\mathbf{u}_k]\!] \rangle - \delta_{jk} \right)^2, \tag{6}$$

where $\boldsymbol{\beta} = [\beta_{1,1}, \beta_{2,1}, \beta_{2,2}, \cdots, \beta_{d,1}, \cdots, \beta_{d,d}] \in \mathbb{R}^{d(d+1)/2}$ is a vector containing all of the dual variables. There are two reasons to introduce the additional linear penalty, which at first glance do not seem to contribute anything that the quadratic one is not adding already. First, for an appropriately chosen $b$, the equilibria of the max-min objective (6) corresponds exactly to permutations of the smallest Laplacian eigenvectors, and only the sorted eigenvectors are a stable solution under gradient ascent-descent dynamics. Second, the optimal dual variables $\boldsymbol{\beta}^*$ are proportional to the smallest Laplacian eigenvalues, meaning that with this single objective one can recover naturally **both** eigenvectors and eigenvalues of $\mathbf{L}$ (see the next section for the formalization of these claims). Moreover, the calculation of the linear penalty is a byproduct of the original quadratic one. This means that the computational complexity is practically unaltered, beyond having to update (and store) $\mathcal{O}(d^2)$ dual parameters, which is negligible considering the number of parameters in neural networks.

Something to note is that the standard augmented Lagrangian has been discussed in the literature as a potential approach for learning eigenvectors of linear operators, but dismissed due to lack of empirical stability (Pfau et al., 2019). **ALLO overcomes this problem** through the introduction of the stop-gradient operators, which are responsible for breaking the symmetry of eigenvector rotations, in a similar way as how gradient masking is used in spectral inference networks (Pfau et al., 2019).

**Barrier Dynamics.** For the introduced max-min objective to work, in theory, $b$ has to be larger than a **finite** value that depends on $\mathbf{L}$. Moreover, if $f(\mathbf{P}_\pi)$ in the definition of $\mathbf{L}$ is a stochastic matrix, which is the case for all definitions mentioned previously, one can exactly determine a lower bound for $b$, as proved in the next section. In practice, however, we found that increasing $b$ is helpful for final performance. In our experiments, we do so in a gradient ascent fashion, just as with the dual variables.

## 4  THEORETICAL RESULTS

To prove the soundness of the proposed max-min objective, we need to show two things: 1) that the equilibria of this objective correspond to the desired eigensystem of the Laplacian, and 2) that this equilibria is stable under stochastic gradient ascent-descent dynamics.

As an initial motivation, the following lemma deals with the first point in the *stationary setting*. While it is known that the solution set for the graph drawing optimization problem (1) corresponds to the rotations of the smallest eigenvectors of $\mathbf{L}$, the Lemma considers a primal-dual perspective of the problem that allows one to relate the dual variables with the eigenvalues of $\mathbf{L}$ (see the Appendix for a proof). This identification is relevant since previous methods are not able to recover the eigenvalues.

**Lemma 1.** *Consider a symmetric matrix $\mathbf{L} \in \mathbb{R}^{|\mathcal{S}| \times |\mathcal{S}|}$ with increasing, and possibly repeated, eigenvalues $\lambda_1 \leq \cdots \leq \lambda_{|\mathcal{S}|}$, and a corresponding sequence of eigenvectors $(\mathbf{e}_i)_{i=1}^{|\mathcal{S}|}$. Then, given a number of components, $1 \leq d \leq |\mathcal{S}|$, the pair $(\mathbf{u}_i^*)_{i=1}^d, (\beta_{jk}^*)_{1 \leq k \leq j \leq d}$, where $\mathbf{u}_i^* = \mathbf{e}_i$ and $\beta_{jk}^* = -\lambda_j \delta_{jk}$, is a solution to the primal-dual pair of optimization problems corresponding to the spectral graph drawing optimization problem (1). Furthermore, any other primal solution corresponds to a rotation of the eigenvectors $(\mathbf{e}_i)_{i=1}^d$.*

Now that we know that the primal-dual pair of optimization problems associated to (1) has as a solution the smallest eigensystem of the Laplacian, the following Lemma shows that the equilibria of the max-min objective (6) coincides only with this solution, up to a constant, and any possible permutation of the eigenvectors, **but not with its rotations**.

**Lemma 2.** *The pair $\mathbf{u}^*, \boldsymbol{\beta}^*$ is an equilibrium pair of the max-min objective (6), under gradient ascent-descent dynamics, if and only if $\mathbf{u}^*$ coincides with a subset of eigenvectors of the Laplacian $(\mathbf{e}_{\sigma(i)})_{i=1}^d$, for some permutation $\sigma : \mathcal{S} \to \mathcal{S}$, and $\beta_{jk}^* = -2\lambda_{\sigma(j)} \delta_{jk}$.*

*Proof.* Denoting $\mathcal{L}$ the objective (6), we have the gradient ascent-descent dynamical system:

$$\mathbf{u}_i[t+1] = \mathbf{u}_i[t] - \alpha_{\text{primal}} \cdot \mathbf{g}_{\mathbf{u}_i}(\mathbf{u}[t], \boldsymbol{\beta}[t]), \ \ \forall 1 \leq i \leq d,$$

$$\beta_{jk}[t+1] = \beta_{jk}[t] + \alpha_{\text{dual}} \cdot \frac{\partial \mathcal{L}}{\partial \beta_{jk}}(\mathbf{u}[t], \boldsymbol{\beta}[t]), \ \ \forall 1 \leq k \leq j \leq d,$$

where $t \in \mathbb{N}$ is the time index, $\alpha_{\text{primal}}, \alpha_{\text{dual}} > 0$ are step sizes, and $\mathbf{g}_{\mathbf{u}_i}$ is the gradient of $\mathcal{L}$ with respect to $\mathbf{u}_i$, considering the stop-gradient operator. We avoid the notation $\nabla_{\mathbf{u}_i} \mathcal{L}$ to emphasize that $\mathbf{g}_{\mathbf{u}_i}$ is not a real gradient, but a direction that ignores what is inside the stop-gradient operator.

The equilibria of our system satisfy $\mathbf{u}_i^*[t+1] = \mathbf{u}_i^*[t]$ and $\beta_{jk}^*[t+1] = \beta_{jk}^*[t]$. Hence,

$$\mathbf{g}_{\mathbf{u}_i}(\mathbf{u}^*, \boldsymbol{\beta}^*) = 2\mathbf{L}\mathbf{u}_i^* + \sum_{j=1}^i \beta_{ij}\mathbf{u}_j^* + 2b \sum_{j=1}^i (\langle \mathbf{u}_i^*, \mathbf{u}_j^* \rangle - \delta_{ij})\mathbf{u}_j^* = \mathbf{0}, \ \ \forall 1 \leq i \leq d, \quad (7)$$

$$\frac{\partial \mathcal{L}}{\partial \beta_{jk}}(\mathbf{u}^*, \boldsymbol{\beta}^*) = \langle \mathbf{u}_j^*, \mathbf{u}_k^* \rangle - \delta_{jk} = 0, \ \ \forall 1 \leq k \leq j \leq d. \quad (8)$$

We proceed now by induction over $i$, considering that Equation (8) tells us that $\mathbf{u}^*$ corresponds to an orthonormal basis. For the base case $i = 1$ we have:

$$\mathbf{g}_{\mathbf{u}_1}(\mathbf{u}^*, \boldsymbol{\beta}^*) = 2\mathbf{L}\mathbf{u}_1^* + \beta_{1,1}\mathbf{u}_1^* = 0.$$

Thus, we can conclude that $\mathbf{u}_1$ is an eigenvector $\mathbf{e}_{\sigma(1)}$ of the Laplacian, and that $\beta_{1,1}$ corresponds to its eigenvalue, specifically $\beta_{1,1} = -2\lambda_{\sigma(1)}$, for some permutation $\sigma : \mathcal{S} \to \mathcal{S}$. Now let us suppose that $\mathbf{u}_j = \mathbf{e}_{\sigma(j)}$ and $\beta_{jk} = -2\lambda_{\sigma(j)}\delta_{jk}$ for $j < i$. Equation (8) for $i$ then becomes:

$$\mathbf{g}_{\mathbf{u}_i}(\mathbf{u}^*, \boldsymbol{\beta}^*) = 2\mathbf{L}\mathbf{u}_i^* + \beta_{ii}\mathbf{u}_i^* + \sum_{j=1}^{i-1} \beta_{ij}\mathbf{e}_{\sigma(j)} = 0.$$

In general, we can express $\mathbf{u}_i^*$ as the linear combination $\mathbf{u}_i^* = \sum_{j=1}^{|\mathcal{S}|} c_{ij}\mathbf{e}_{\sigma(j)}$ since the eigenvectors of the Laplacian form a basis. Also, given that $\langle \mathbf{u}_j, \mathbf{u}_k \rangle = 0$, we have that $c_{ij} = 0$ for $j < i$. Hence,

$$2\sum_{j=i}^{|\mathcal{S}|} c_{ij}\mathbf{L}\mathbf{e}_{\sigma(j)} + \beta_{ii} \sum_{j=i}^{|\mathcal{S}|} c_{ij}\mathbf{e}_{\sigma(j)} + \sum_{j=1}^{i-1} \beta_{ij}\mathbf{e}_{\sigma(j)} = \sum_{j=i}^{|\mathcal{S}|} c_{ij}(2\lambda_{\sigma(j)} + \beta_{ii})\mathbf{e}_{\sigma(j)} + \sum_{j=1}^{i-1} \beta_{ij}\mathbf{e}_{\sigma(j)} = 0.$$

By orthogonality of the eigenvectors, we must have that each coefficient is 0, implying that $\beta_{ij} = 0$ and either $c_{ij} = 0$ or $\beta_{ii} = -2\lambda_{\sigma(j)}$. The last equation allows us to conclude that a pair $(c_{ij}, c_{ik})$ can only be different to 0 simultaneously for $j, k$ such that $\lambda_{\sigma(j)} = \lambda_{\sigma(k)}$, i.e., $\mathbf{u}_i$ lies in the subspace of eigenvectors corresponding to the same eigenvalue, where each point is in itself an eigenvector. Thus, we can conclude, that $\mathbf{u}_i = \mathbf{e}_{\sigma(i)}$ and $\beta_{ij} = -2\lambda_i\delta_{ij}$, as desired. □

As a corollary to Lemma 2, let us suppose that we fix all the dual variables to 0, i.e., $\beta_{jk} = 0$. Then, we will obtain that the constraints of the original optimization problem (1) must be violated for any possible equilibrium point (see the Appendix for a proof). This explains why optimizing GGDO in Equation (4) may converge to undesirable rotations of the Laplacian eigenvectors, even when the smallest eigenvectors are the unique solution of the original constrained optimization problem.

**Corollary 1.** *The point $\mathbf{u}^*$ is an equilibrium point of objectives (2) or (4), under gradient descent dynamics, if and only if, for any $1 \le i \le d$, there exists a $1 \le j \le d$ such that $\langle \mathbf{u}_i^*, \mathbf{u}_j^* \rangle \neq \delta_{ij}$. That is, the equilibrium is guaranteed to be different to the eigenvectors of the Laplacian.*

Finally, we prove that even when all permutations of the Laplacian eigenvectors are equilibrium points of the proposed objective (6), only the one corresponding to the ordered smallest eigenvectors and its eigenvalues is stable (see Appendix D for empirical confirmation), in contrast with GGDO.

**Theorem 1.** *The only permutation in Lemma 2 that corresponds to an stable equilibrium point of the max-min objective (6) is the identity permutation, under an appropriate selection of the barrier coefficient $b$, if the highest eigenvalue multiplicity is 1. That is, there exist a finite barrier coefficient such that $\mathbf{u}_i^* = \mathbf{e}_i$ and $\beta_{jk}^* = -2\lambda_j\delta_{jk}$ correspond to the only stable equilibrium pair, where $\lambda_i$ is the $i-$th smallest eigenvalue of the Laplacian and $\mathbf{e}_i$ its corresponding eigenvector. In particular, any $b > 2$ guarantees stability.*

*Proof Sketch.* Complete proof is in the Appendix. We have that $\mathbf{g}_{\mathbf{u}_i}$ and $\partial\mathcal{L}/\partial\beta_{jk}$ define the chosen ascent-descent direction. Concatenating these vectors and scalars in a single vector $\mathbf{g}(\mathbf{u}, \boldsymbol{\beta})$, the stability of the dynamics can be determined from the Jacobian matrix $J(\mathbf{g})$. Specifically, if all the eigenvalues of this matrix have a positive real part in the equilibrium pair $\mathbf{u}^*, \boldsymbol{\beta}^*$, we can conclude that the equilibrium is stable. If there is one eigenvalue with negative real part, then it is unstable (see Chicone, 2006; Sastry, 2013; Mazumdar et al., 2020). As proved in the Appendix, for any pair $1 \le i < j \le |\mathcal{S}|$, there exists a real eigenvalue proportional to $\lambda_{\sigma(j)} - \lambda_i$. This means that, unless the $\sigma$ permutation is the identity, there will be at least one negative eigenvalue and the equilibrium corresponding to this permutation will be unstable. □

## 5 EXPERIMENTS

We evaluate three different aspects of the proposed max-min objective: eigenvector accuracy, eigenvalue accuracy, and the necessity of each of the components of the proposed objective.[6]

**Eigenvector Accuracy.** We start by considering the grid environments shown in Figure 2. We generate $200,000$ transition samples in each of them from a uniform random policy and a uniform initial state distribution. We use the $(x, y)$ coordinates as inputs to a fully-connected neural network $\phi_{\boldsymbol{\theta}} : \mathbb{R}^2 \to \mathbb{R}^d$, parameterized by $\boldsymbol{\theta}$, with 3 layers of 256 hidden units to approximate the $d-$dimensional Laplacian representation $\phi$, where $d = 11$. The network is trained using stochastic gradient descent with our objective (see Wu et al., 2019), for the same initial barrier coefficients as in Figure 1.

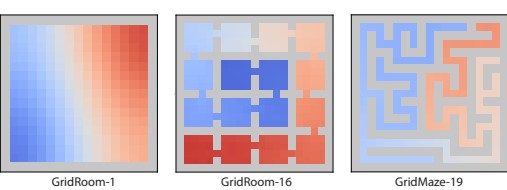

GridRoom-1    GridRoom-16    GridMaze-19

Figure 2: Grid environments. Color is the 2nd smallest Laplacian eigenvector learned by ALLO.

Figure 3 shows the average cosine similarity of eigenvectors found using ALLO compared to the true Laplacian eigenvectors. In all three environments, it learns close approximations of the smallest

---

[6]Accompanying code is available here: `https://github.com/tarod13/laplacian_dual_dynamics`.

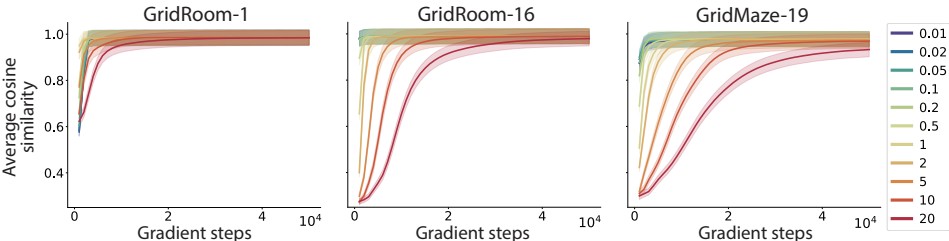

Figure 3: Average cosine similarity between the true Laplacian and ALLO for different initial values of the barrier coefficient $b$, averaged over 60 seeds, with the best coefficient highlighted. The shaded region corresponds to a 95% confidence interval.

$d-$eigenvectors in fewer gradient updates than GGDO (see Figure 1) and without a strong dependence on the chosen barrier coefficients.

As a second and more conclusive experiment, we select the barrier coefficient with the best performance for GGDO across the three previous environments ($b = 2.0$), and the best barrier increasing rate, $\alpha_{\text{barrier}}$, for our method across the same environments ($\alpha_{\text{barrier}} = 0.01$). Then, we use these values to learn the Laplacian representation in 12 different grid environments, each with different number of states and topology (see Figure 6 in the Appendix), and we consider both the $(x, y)$ and pixel-based representations (each tile of the grid corresponds to a pixel), with 1 million samples.

Figure 4A compares the average cosine similarities obtained with each method for the $(x, y)$ representation (see detailed results for both representations in Appendix E). In particular, it shows the mean difference of the average cosine similarities across 60 seeds. Noticeably, the baseline fails completely in the two smallest environments (i.e., `GridMaze-7` and `GridMaze-9`), and it also fails partially in the two largest ones (i.e., `GridMaze-32` and `GridRoom-64`). In contrast, ALLO finds close approximations of the true Laplacian representation across all environments, with the exception of `GridRoomSym-4`, where it still found a more accurate representation than GGDO. These results are statistically significant for 9 out of 12 environments, with a p-value threshold of 0.01 (see Table 1 in the Appendix). Again, this suggests that the proposed objective is successful in removing the untunable-hyperparameter dependence observed in GGDO.

**Eigenvalue Accuracy.** The dual variables of ALLO should capture the eigenvalues of their associated eigenvectors. Here, we quantify how well they approximate the true eigenvalues in the same 12 grid environments as in Figure 4A. In particular, we compare our eigenvalue accuracy against those found with a simple alternative method (Wang et al., 2023), based on GGDO and on Monte Carlo approximations. Figure 4B shows that the average relative error for the second to last eigenvalues, meaning all except one, is consistently larger across all environments when using the alternative approach, with a significance level of 0.01. This is not surprising given the poor results in eigenvector accuracy for GGDO. However, in several environments the error is high even for the smallest eigenvalues, despite GGDO approximations being relatively more accurate for the associated eigenvectors. Across environments and across the eigenspectrum, our proposed objective provides more accurate estimates of the eigenvalues (see Appendix F for the exact values).

**Ablations.** ALLO has three components that are different from GGDO: (1) the **stop-gradient** as a mechanism to break the symmetry, (2) the **dual variables** that penalize the linear constraints and from which we extract the eigenvalues of the graph Laplacian, and (3) the mechanism to monotonically **increase the barrier coefficient** that scales the quadratic penalty. Our theoretical results suggest that the stop-gradient operation and the dual variables are necessary, while increasing the barrier coefficient could be helpful, eventually eliminating the need for the dual variables if all one cared about was to approximate the eigenvectors of the graph Laplacian, not its eigenvalues. In this section, we perform ablation studies to validate whether these insights translate into practice when using neural networks to minimize our objective. Specifically, in `GridMaze-19`, we compare the average cosine similarity of ALLO, with the same objective but without dual variables, and with GGDO, which does not use dual variables, nor the stop gradient, nor the increasing coefficients. For completeness, we also evaluate GGDO objective with increasing coefficients.

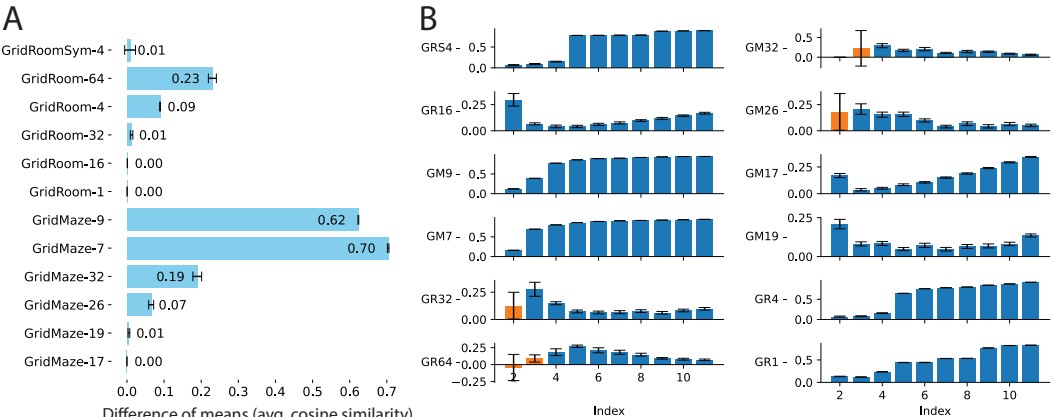

Figure 4: Difference of cosine similarities when approximating eigenvectors (A), and of relative errors for eigenvalues (B). Error bars show the standard deviation of the differences. `GR` and `GM` stand for `GridRoom` and `GridMaze`. black bars correspond to p-values below $0.01$.

The curves in each panel of Figure 5 represent the different methods we evaluate, while the different panels evaluate the impact of different rates of increase of the barrier coefficient. Our results show that increasing the barrier coefficients is indeed important, and not increasing it (as in GDO and GGDO) actually prevents us from obtaining the true eigenvectors. It is also interesting to observe that the rate in which we increase the barrier coefficient matters empirically, but it does not prevent our solution to obtain the true eigenvectors. The importance of the stop gradient is evident when one looks at the difference in performance between GGDO and ALLO (and variants), particularly when not increasing the barrier coefficients. Finally, it is interesting to observe that the addition of the dual variables, which is essential to estimate the eigenvalues of the graph Laplacian, does not impact the performance of our approach. Based on our theoretical results, we conjecture the dual variables add stability to the learning process in larger environments, but we leave this for future work.

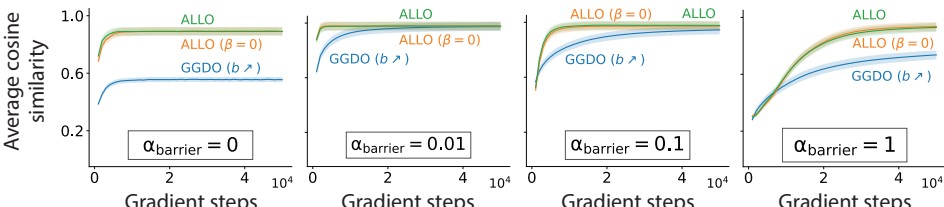

Figure 5: Average cosine similarity for different objectives in the environment `GridMaze-19`, for initial barrier coefficient $b = 0.1$, and for different barrier increase rates $\alpha_{\text{barrier}}$.

## 6 CONCLUSION

In this paper we introduced a theoretically sound min-max objective that makes use of stop-gradient operators to turn the Laplacian representation into the unique stable equilibrium point of a gradient ascent-descent optimization procedure. We showed empirically that, when applied to neural networks, the objective is robust to the same untunable hyperparameters that affect alternative objectives across environments with diverse topologies. In addition, we showed how the objective results in a more accurate estimation of the Laplacian eigenvalues when compared to alternatives.

As future work, it would be valuable to better understand the theoretical impact of the barrier coefficient in the optimization process. Since we can now obtain the eigenvalues of the graph Laplacian, it would be also interesting to see how they could be leveraged, e.g., as an emphasis vector for feature representations or as a proxy for the duration of temporally-extended actions discovered from the Laplacian. Finally, it would be exciting to see the impact that having access to a proper approximation of the Laplacian will have in algorithms that rely on it (e.g., Wang et al., 2023; Klissarov & Machado, 2023).

ACKNOWLEDGMENTS

We thank Alex Lewandowski for helpful discussions about the Laplacian representation, Martin Klissarov for providing an initial version of the baseline (GGDO), and Adrian Orenstein for providing useful references on augmented Lagrangian techniques. The research is supported in part by the Natural Sciences and Engineering Research Council of Canada (NSERC), the Canada CIFAR AI Chair Program, and the Digital Research Alliance of Canada.

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

## A  ADDITIONAL THEORETICAL DERIVATIONS

### A.1  PROOF OF LEMMA 1

*Proof.* Let $\beta_{jk} \in \mathbb{R}$ denote the dual variables associated to the constraints of the optimization problem (1), and $\mathcal{L}$ be the corresponding Lagrangian function:

$$\mathcal{L}((\mathbf{u}_i)_i, (\beta_{jk})_{k \leq j}) := \sum_{i=1}^{d} \langle \mathbf{u}_i, \mathbf{L}\mathbf{u}_i \rangle + \sum_{j=1}^{d} \sum_{k=1}^{j} \beta_{jk}(\langle \mathbf{u}_j, \mathbf{u}_k \rangle - \delta_{jk}).$$

Then, any pair of solutions $(\mathbf{u}_i^*)_i, (\beta_{jk}^*)_{k \leq j}$ must satisfy the Karush-Kuhn-Tucker conditions. In particular, the gradient of the Lagrangian should be 0 for both primal and dual variables:

$$\nabla_{\mathbf{u}_i} \mathcal{L}((\mathbf{u}_i^*)_i, (\beta_{jk}^*)_{k \leq j}) = 2\mathbf{L}\mathbf{u}_i^* + 2\beta_{ii}^* \mathbf{u}_i^* + \sum_{k=1}^{i-1} \beta_{ik}^* \mathbf{u}_k^* + \sum_{j=i+1}^{d} \beta_{ji}^* \mathbf{u}_j^* = 0, \ \ \forall 1 \leq i \leq d; \quad (9)$$

$$\nabla_{\beta_{jk}} L((\mathbf{u}_i^*)_i, (\beta_{jk}^*)_{k \leq j}) = \langle \mathbf{u}_j^*, \mathbf{u}_k^* \rangle - \delta_{jk} = 0, \ \ \forall 1 \leq k \leq j \leq d. \quad (10)$$

The Equation (10) does not introduce new information since it only asks again for the solution set $(\mathbf{u}_i^*)_{i=1}^{d}$ to form an orthonormal basis. Equation (9) is telling us something more interesting. It asks $\mathbf{L}\mathbf{u}_i^*$ to be a linear combination of the vectors $(\mathbf{u}_i^*)_{i=1}^{d}$, i.e., it implies that $\mathbf{L}$ always maps $\mathbf{u}_i^*$ back to the space spanned by the basis. Since this is true for all $i$, the span of $(\mathbf{u}_i^*)_{i=1}^{d}$ must coincide with the span of the eigenvectors $(\mathbf{e}_{\sigma(i)})_{i=1}^{d}$, for some permutation $\sigma : \mathcal{S} \to \mathcal{S}$, as proved in Proposition 1, also in the Appendix. Intuitively, if this was not the case, then the scaling effect of $\lambda_j$ along some $\mathbf{e}_{\sigma(j)}$ would take points that are originally in $\text{span}(\mathbf{u}_i^*)_{i=1}^{d}$ outside of this hyperplane.

Since we know that the span of the desired basis is $\text{span}(\mathbf{e}_{\sigma(i)})_{i=1}^{d}$, for some permutation $\sigma : \mathcal{S} \to \mathcal{S}$, we can restrict the solution to be a set of eigenvectors of $\mathbf{L}$. The function being minimized then becomes $\sum_{i=1}^{d} \lambda_{\sigma(i)}$, which implies that a primal solution is the set of $d$ smallest eigenvectors. Now, any rotation of this minimizer results in the same loss and is also in $\text{span}(\mathbf{e}_i))_{i=1}^{d}$, which implies that any rotation of these eigenvectors is also a primal solution.

Considering the primal solution where $\mathbf{u}_i^* = \mathbf{e}_i$, Equation (9) becomes:

$$\nabla_{\mathbf{u}_i} \mathcal{L}((\mathbf{e}_i)_i, (\beta_{jk}^*)_{k \leq j}) = 2(\lambda_i + \beta_{ii}^*)\mathbf{e}_i + \sum_{k=1}^{i-1} \beta_{ik}^* \mathbf{e}_k + \sum_{j=i+1}^{d} \beta_{ji}^* \mathbf{e}_j = 0, \ \ \forall 1 \leq i \leq d.$$

Since the eigenvectors are normal to each other, the coefficients all must be 0, which implies that the corresponding dual solution is $\beta_{ii}^* = -\lambda_i$ and $\beta_{jk}^* = 0$ for $j \neq k$. $\qquad\square$

## A.2 PROPOSITION 1

**Proposition 1.** *Let* $\mathbf{T} \in \mathbb{R}^{S \times S}$ *be a symmetric matrix,* $\mathbf{u}_1, \cdots, \mathbf{u}_S \in \mathbb{R}^S$ *and* $\lambda_1, \cdots, \lambda_S \in \mathbb{R}$ *be its eigenvectors and corresponding eigenvalues, and* $\mathbf{e}_1, \cdots, \mathbf{e}_d \in \mathbb{R}^S$ *be a* $d-$*dimensional orthonormal basis of the subspace* $\mathcal{E} := span((\mathbf{e}_i)_i)$. *Then, if* $\mathcal{E}$ *is closed under the operation of* $\mathbf{T}$*, i.e.,* $\mathbf{T}(\mathcal{E}) \subseteq \mathcal{E}$*, there must exist a* $d-$*dimensional subset of eigenvectors* $\{\mathbf{v}_1, \cdots, \mathbf{v}_d\} \subseteq \{\mathbf{u}_1, \cdots, \mathbf{u}_S\}$ *such that* $\mathcal{E}$ *coincides with their span, i.e.,* $\mathcal{E} = span((\mathbf{v}_i)_i) = span((\mathbf{e}_i)_i)$.

*Proof.* Let $\mathbf{w} \neq 0$ be a vector in $\mathcal{E}$. Then, by definition of $\mathcal{E}$, it can be expressed as a linear combination $\mathbf{w} = \sum_{i=1}^d \alpha_i \mathbf{e}_i$, where $\alpha_i \in \mathbb{R}$, and at least one of the coefficients is non-zero. Let us consider now the operation of $\mathbf{T}$ on $\mathbf{w}$ in terms of its eigenvectors. Specifically, we can express it as

$$\mathbf{Tw} = \sum_{j=1}^S \lambda_j \langle \mathbf{u}_j, \mathbf{w} \rangle \mathbf{u}_j \,,$$

which, by linearity of the inner-product, becomes

$$\mathbf{Tw} = \sum_{j=1}^S \left( \lambda_j \sum_{i=1}^d \alpha_i \langle \mathbf{u}_j, \mathbf{e}_i \rangle \right) \mathbf{u}_j \,.$$

Considering the hypothesis that $\mathcal{E}$ is closed under $\mathbf{T}$, we reach a necessary condition:

$$\sum_{j=1}^S \left( \lambda_j \sum_{i=1}^d \alpha_i \langle \mathbf{u}_j, \mathbf{e}_i \rangle \right) \mathbf{u}_j \overset{!}{=} \sum_{i=1}^d \beta_i \mathbf{e}_i \,, \tag{11}$$

where $\beta_i \in \mathbb{R}$, and at least one of them is non-zero.

We proceed by contradiction. Let us suppose that there does not exist a $d-$dimensional subset of eigenvectors $\{\mathbf{v}_1, \cdots, \mathbf{v}_d\}$ such that $\mathcal{E} = \text{span}((\mathbf{v}_i)_i)$. Since the eigenvectors form a basis of the whole space, we can express each $\mathbf{e}_i$ as linear combinations of the form

$$\mathbf{e}_i = \sum_{j=1}^S c_{ij} \mathbf{u}_i = \sum_{j=1}^S \langle \mathbf{u}_j, \mathbf{e}_i \rangle \mathbf{u}_j \,.$$

So, supposing that $\mathcal{E}$ does not correspond to any eigenvector subspace, there must exist $d' > d$ *different* indices $j_1, \cdots, j_{d'}$ and corresponding pairs $(i_k, j_k)$ such that $c_{i_k j_k} \neq 0$. If this was not the case, this would imply that all the $\mathbf{e}_i$ lie in the span of some subset of $d$ or less eigenvectors, and so $\mathcal{E}$ would correspond to this span.

Hence, we have that the coefficients $\alpha_i$ are arbitrary and that at least $d+1$ inner products are not zero. This implies that $\mathbf{w}$ lies in a subspace of dimension at least $d+1$ spanned by the $d'$ eigenvectors $\mathbf{v}_1, \cdots, \mathbf{v}_{d'}$ with $\mathbf{v}_k = \mathbf{u}_{j_k}$. Now, the condition in Equation (11) requires this subspace to be the same as $\mathcal{E}$, but this is not possible since $\mathcal{E}$ is $d-$dimensional. Thus, we can conclude that there must exist a basis of $d$ eigenvectors $\mathbf{v}_1, \cdots, \mathbf{v}_d$ of $\mathbf{T}$ such that $\mathcal{E} = \text{span}((\mathbf{v}_i)_i) = \text{span}((\mathbf{e}_i)_i)$. $\qquad\square$

## A.3 PROOF OF COROLLARY 1

*Proof.* Taking the gradient of GGDO (4), referred to as $\mathcal{L}$ here, leads us to a similar expression as in Equation (7) for a pair $(\mathbf{u}, \boldsymbol{\beta} = \mathbf{0})$ and index $i$:

$$\nabla_{\mathbf{u}_i} \mathcal{L}(\mathbf{u}) = 2c_i \mathbf{L} \mathbf{u}_i + \underbrace{\sum_{j=1}^d \beta_{ij} \mathbf{u}_j}_{=\mathbf{0}} + 2b \sum_{j=1}^d c_{ij} (\langle \mathbf{u}_i, \mathbf{u}_j \rangle - \delta_{ij}) \mathbf{u}_j \,,$$

where $c_{ij}$ is some constant function of $c_i$ and $c_j$. Comparing with the proof of Lemma 1 in A.1, we can notice that there are two key differences: the dual sum $\sum_{j=1}^{d} \beta_{ij} \mathbf{u}_j$ is now $\mathbf{0}$, and the indices in the barrier term $\sum_{j=1}^{d} c_{ij}(\langle \mathbf{u}_i, \mathbf{u}_j \rangle - \delta_{ij})\mathbf{u}_j$ go now from $1$ to $d$, as opposed to only $1$ to $i$. If we require the gradient to be $\mathbf{0}$ as well, meaning that $\mathbf{u}$ is an equilibrium point, then we have that:

$$\mathbf{L}\mathbf{u}_i = \sum_{j=1}^{d} C_{ij}(\langle \mathbf{u}_i, \mathbf{u}_j \rangle - \delta_{ij})\mathbf{u}_j \,,$$

for some non-zero constants $C_{ij}$. That is, the vector $\mathbf{L}\mathbf{u}_i$ has to lie in the subspace spanned by the $d$ vectors $\mathbf{u}_1, \cdots, \mathbf{u}_d$. This is only possible, however, if at least 1 of the coefficients $\langle \mathbf{u}_i, \mathbf{u}_j \rangle - \delta_{ij}$ is different from 0 for some $j$. But then, this means that the restrictions of the original optimization problem are violated in any possible equilibrium point. $\qquad\square$

### A.4 PROOF OF THEOREM 1

*Proof.* Let us define the following vectors defining the descent directions for $\mathbf{u}$ and $\boldsymbol{\beta}$:

$$\mathbf{g_u} = \begin{bmatrix} \mathbf{g_{u_1}} \\ \mathbf{g_{u_2}} \\ \vdots \\ \mathbf{g_{u_d}} \end{bmatrix}, \quad \mathbf{g_\beta} = \begin{bmatrix} \frac{\partial \mathcal{L}}{\partial \beta_{1,1}} \\ \frac{\partial \mathcal{L}}{\partial \beta_{2,1}} \\ \frac{\partial \mathcal{L}}{\partial \beta_{2,2}} \\ \vdots \\ \frac{\partial \mathcal{L}}{\partial \beta_{d,1}} \\ \vdots \\ \frac{\partial \mathcal{L}}{\partial \beta_{d,d}} \end{bmatrix}.$$

Then, the global ascent-descent direction can be represented by the vector

$$\mathbf{g} = \begin{bmatrix} \mathbf{g_u} \\ -\mathbf{g_\beta} \end{bmatrix}.$$

To determine the stability of any equilibrium point of the ascent-descent dynamics introduced in Lemma 2, we only need to calculate the Jacobian of $\mathbf{g}$, the matrix $\mathbf{J} := J(\mathbf{g})$ whose rows correspond to the gradients of each entry of $\mathbf{g}$, and determine its eigenvalues (Chicone, 2006).

We proceed to take the gradients of Equation 7 and 8:

$$\mathbf{J}_{ij}(\mathbf{u}, \boldsymbol{\beta}) := (\nabla_{\mathbf{u}_i} \mathbf{g}_{\mathbf{u}_j}(\mathbf{u}, \boldsymbol{\beta})^\top)^\top \tag{12}$$

$$= \begin{cases} 2\mathbf{L} + \beta_{ii}\mathbf{I} + 2b\big[(\langle \mathbf{u}_i, \mathbf{u}_i \rangle - 1)\mathbf{I} + 2\mathbf{u}_i \otimes \mathbf{u}_i\big] + 2\sum_{k=1}^{i-1} b\mathbf{u}_k \otimes \mathbf{u}_k, & \text{if } i = j\,; \\[2mm] \beta_{ij}\mathbf{I} + 2b\big(\langle \mathbf{u}_i, \mathbf{u}_j \rangle \mathbf{I} + \mathbf{u}_i \otimes \mathbf{u}_j\big), & \text{if } i > j\,; \\[2mm] \mathbf{0}, & \text{if } i < j\,; \end{cases}$$

$$\mathbf{J}_{k\beta_{ij}}(\mathbf{u}, \boldsymbol{\beta}) := \left( -\nabla_{\mathbf{u}_k} \frac{\partial \mathcal{L}}{\partial \beta_{ij}}(\mathbf{u}, \boldsymbol{\beta}) \right)^\top = -\mathbf{u}_j^\top \delta_{ik} - \mathbf{u}_i^\top \delta_{jk}\,; \tag{13}$$

$$\mathbf{J}_{\beta_{jk}i}(\mathbf{u}, \boldsymbol{\beta}) := \frac{\partial}{\partial \beta_{jk}} \mathbf{g}_{\mathbf{u}_i}(\mathbf{u}, \boldsymbol{\beta}) = 2\mathbf{u}_i \delta_{ij}\delta_{ik} + \mathbf{u}_k \delta_{ij}(1 - \delta_{jk})\,; \tag{14}$$

$$\mathbf{J}_{\beta_{k\ell}\beta_{ij}}(\mathbf{u}, \boldsymbol{\beta}) := \frac{\partial^2 \mathcal{L}}{\partial \beta_{k\ell}\partial \beta_{ij}}(\mathbf{u}, \boldsymbol{\beta}) = 0\,. \tag{15}$$

Then, we have that in any equilibrium point $\mathbf{u}^*, \boldsymbol{\beta}^*$, i.e., in a permutation $\sigma$ of the Laplacian eigensystem (as per Lemma 2), the Jacobian satisfies:

$$
\mathbf{J}_{ij}(\mathbf{u}^*, \boldsymbol{\beta}^*) = \begin{cases} 2\mathbf{L} - 2\lambda_i\mathbf{I} + 4b\mathbf{e}_{\sigma(i)} \otimes \mathbf{e}_{\sigma(i)} + 2\sum_{k=1}^{i-1} b\mathbf{e}_{\sigma(k)} \otimes \mathbf{e}_{\sigma(k)}, & \text{if } i = j; \\ 2b\mathbf{e}_{\sigma(i)} \otimes \mathbf{e}_{\sigma(j)}, & \text{if } i > j; \\ \mathbf{0}, & \text{if } i < j; \end{cases}
$$

$$
\mathbf{J}_{k\beta_{ij}}(\mathbf{u}^*, \boldsymbol{\beta}^*) = -\mathbf{e}_{\sigma(j)}^\top \delta_{ik} - \mathbf{e}_{\sigma(i)}^\top \delta_{jk};
$$

$$
\mathbf{J}_{\beta_{jk}i}(\mathbf{u}^*, \boldsymbol{\beta}^*) = 2\mathbf{e}_{\sigma(i)}\delta_{ij}\delta_{ik} + \mathbf{e}_{\sigma(k)}\delta_{ij}(1 - \delta_{jk});
$$

$$
\mathbf{J}_{\beta_{k\ell}\beta_{ij}}(\mathbf{u}^*, \boldsymbol{\beta}^*) = 0.
$$

Now, we determine the eigenvalues of this Jacobian. For this, we need to solve the system:

$$
\mathbf{J}\mathbf{v} = \eta\mathbf{v}, \tag{16}
$$

where $\eta$ denotes an eigenvalue of the Jacobian and $\mathbf{v}$, its corresponding eigenvector.

To facilitate the solution of this system, we use the following notation:

$$
\mathbf{v} = \begin{bmatrix} \mathbf{w} \\ \boldsymbol{\nu} \end{bmatrix}, \quad \mathbf{v_u} = \begin{bmatrix} \mathbf{w}_1 \\ \mathbf{w}_2 \\ \vdots \\ \mathbf{w}_d \end{bmatrix}, \quad \boldsymbol{\nu} = \begin{bmatrix} \nu_{1,1} \\ \nu_{2,1} \\ \nu_{2,2} \\ \vdots \\ \nu_{d,1} \\ \vdots \\ \nu_{d,d} \end{bmatrix},
$$

where $\mathbf{w}_i \in \mathbb{R}^{|\mathcal{S}|}$, for all $1 \le i \le d$, and $\nu_{jk} \in \mathbb{R}$, for all $1 \le k \le j \le d$. With this, the eigenvalue system (16) becomes:

$$
\begin{cases} \sum_{j=1}^d \mathbf{J}_{ji}\mathbf{w}_j + \sum_{j=1}^d \sum_{k=1}^j \mathbf{J}_{\beta_{jk}i}\nu_{jk} = \eta\mathbf{w}_i & \forall\, 1 \le i \le d \\ \sum_{k=1}^d \mathbf{J}_{k\beta_{ij}}\mathbf{w}_k = \eta\nu_{ij} & \forall\, 1 \le j \le i \le d \end{cases}. \tag{17}
$$

Since the Laplacian eigenvectors form a basis, we have the decomposition $\mathbf{w}_i = \sum_{j=1}^{|\mathcal{S}|} c_{ij}\mathbf{e}_{\sigma(j)}$, for some sequence of reals $(c_{ij})_{j=1}^{|\mathcal{S}|}$. Hence, replacing the values of the Jacobian components in the upper equation of the system (17), we obtain:

$$
\sum_{j=1}^{i-1} 2b(\mathbf{e}_{\sigma(i)} \otimes \mathbf{e}_{\sigma(j)})\mathbf{w}_j + \left(2\mathbf{L} - 2\lambda_i\mathbf{I} + 4b\mathbf{e}_{\sigma(i)} \otimes \mathbf{e}_{\sigma(i)} + 2\sum_{k=1}^{i-1} b\mathbf{e}_{\sigma(k)} \otimes \mathbf{e}_{\sigma(k)}\right)\mathbf{w}_i + \cdots
$$

$$
\cdots + \sum_{j=1}^d \sum_{k=1}^j \left(2\mathbf{e}_{\sigma(i)}\delta_{ij}\delta_{ik} + \mathbf{e}_{\sigma(k)}\delta_{ij}(1 - \delta_{jk})\right)\nu_{jk} - \eta\mathbf{w}_i = 0
$$

$$
\implies 2\sum_{j=1}^{i-1} bc_{ji}\mathbf{e}_{\sigma(j)} + 2\sum_{j=1}^{|\mathcal{S}|}(\lambda_j - \lambda_i)c_{ij}\mathbf{e}_{\sigma(j)} + 4b\mathbf{e}_{\sigma(i)} + \sum_{j=1}^{i-1} bc_{ij}\mathbf{e}_{\sigma(j)} + \cdots
$$

$$\cdots + 2\nu_{ii}\mathbf{e}_{\sigma(i)} + \sum_{k=1}^{i-1} \nu_{ik}\mathbf{e}_{\sigma(k)} - \eta \sum_{j=1}^{|\mathcal{S}|} c_{ij}\mathbf{e}_{\sigma(j)} = 0\,.$$

Since the eigenvectors form a basis, we have that each coefficient in the sum of terms we have must be 0. Hence, we obtain the following conditions:

$$\begin{cases} c_{ij}[2(\lambda_{\sigma(j)} - \lambda_i) + 2b - \eta] = 2bc_{ji} - \nu_{ij}\,, & \forall\, 1 \le j < i \le d \\[2mm] c_{ii}(4b - \eta) = -2\nu_{ii}\,, & \forall\, 1 \le i \le d \\[2mm] c_{ij}[2(\lambda_{\sigma(j)} - \lambda_i) - \eta] = 0\,, & \forall\, 1 \le i < j \le |\mathcal{S}| \end{cases} \tag{18}$$

Each of these conditions specify the possible eigenvalues of the Jacobian matrix $\mathbf{J}$. First and foremost, the third condition tells us that $\eta = 2(\lambda_{\sigma(j)} - \lambda_i)$ is an eigenvalue independent of $b$, for any possible pair $i \le j$. Since we are supposing the eigenvalues are increasing with their index, for the eigenvalues to be positive, the permutation $\sigma : \mathcal{S} \to \mathcal{S}$ must preserve the order for all indexes, which only can be true for the identity permutation. That is, all the Laplacian eigenvector permutations that are not sorted are unstable.[7]

In addition, deriving the rest of the eigenvalues from the remaining two conditions in (18) and the second set of equations of the system (17), we obtain a lower bound for $b$ that guarantees the stability of the Laplacian representation. In particular, from the second set of equations of the system (17) we can obtain a relationship between the coefficients $c_{ij}$ and $c_{ji}$ with $\nu_{ij}$, for all $1 \le j \le i \le d$:

$$\sum_{k=1}^{d} \mathbf{J}_{j\beta_{ij}}\mathbf{w}_k = \sum_{k=1}^{d} \left( -\mathbf{e}_{\sigma(j)}^{\top}\delta_{ik} - \mathbf{e}_{\sigma(i)}^{\top}\delta_{jk} \right)\left( \sum_{\ell=1}^{|\mathcal{S}|} c_{k\ell}\mathbf{e}_{\sigma(\ell)} \right) = -c_{ij} - c_{ji} = \eta\nu_{ij}\,. \tag{19}$$

Replacing this into the second condition in (18) we get that $\eta = 2b \pm 2\sqrt{b^2 - 1}$. These set of eigenvalues (two for each $i$) always have a positive real part, as long as $b$ is strictly positive. In addition, if $b \ge 1$, we get purely real eigenvalues, which are associated with a less oscillatory behavior (see (Sastry, 2013)).

Finally, if we assume that $\eta \ne 2(\lambda_{\sigma(i)} - \lambda_j)$ for $j < i$ (i.e., $\eta$ is not an eigenvalue already considered), we must have that $c_{ji} = 0$, and so, by (19), $-c_{ij} = \eta\nu_{ij}$. Replacing this into the first condition in (18), we get that $\eta = (\lambda_{\sigma(j)} - \lambda_i) + b \pm \sqrt{[(\lambda_{\sigma(j)} - \lambda_i) + b]^2 - 1} \ge -2 + b - \sqrt{[(\lambda_{\sigma(j)} - \lambda_i) + b]^2 - 1}$. Thus, if $b$ is larger than the maximal eigenvalue difference for the first $d$ eigenvalues of $\mathbf{L}$, we have guaranteed that these eigenvalues of the Jacobian will be positive. Furthermore, since the eigenvalues are restricted to the range $[0, 2]$, we have that $b > 2$ ensures a strict stability of the Laplacian representation. □

### A.5 ABSTRACT SETTING

For sake of completeness, we now discuss how Lemma 2 and Theorem 1 are affected when we consider an abstract measure state space.

**Abstract measure state spaces.** First, our state space is now a triple $(\mathcal{S}, \Sigma, \rho)$, where $\mathcal{S}$ is a set of states, $\Sigma$ is an appropriate $\sigma-$algebra contained in the power set $\mathcal{P}(\mathcal{S})$ and whose elements are those sets of states that can be measured, and $\rho : \Sigma \to [0, 1]$ is a valid probability measure for the pair $(\mathcal{S}, \Sigma)$. Also, for clarity, let us denote now the state distribution $P(s, a)$ corresponding to the state-action pair $(s, a)$ as $\mathbb{P}(\cdot|s, a)$. In this manner, while the policy $\pi$ remains the same, the Markov process induced by it cannot be represented anymore by a matrix $\mathbf{P}_\pi$. In its place, we have now the transition probability map $\mathbb{P}_\pi : \mathcal{S} \to \Delta(\mathcal{S})$,[8] defined by $\mathbb{P}_\pi(\cdot|s) = \sum_{a \in \mathcal{A}} \pi(a|s)\mathbb{P}(\cdot|s, a)$.

---

[7]Note that this conclusion only apply to the case where there are no repeated eigenvalues. This is the case since otherwise $\eta = 0$ is an eigenvalues and then the Hartman-Grobman theorem that allows to conclude stability is not conclusive.

[8]The simplex $\Delta(\mathcal{S})$ refers here to the set of probability measures defined over $(\mathcal{S}, \Sigma)$.

Moreover, associated with each transition probability measure $\mathbb{P}_\pi(\cdot|s)$ for $s \in \mathcal{S}$, there is a transition probability density, $p_\pi(\cdot|s) : \mathcal{S} \to [0, \infty)$, defined as the Radon-Nikodym derivative $p_\pi(s'|s) = \frac{d\mathbb{P}_\pi(\cdot|s)}{d\rho}\big|_{s'}$. This is the same to say that the probability of reaching a state in the set $\mathcal{B} \subset \mathcal{S}$, starting from state $s$, is the integral of the density in $\mathcal{B}$: $\mathbb{P}_\pi(\mathcal{B}|s) = \int_\mathcal{B} p_\pi(s'|s)\rho(ds')$.

**Abstract Laplacian.** Second, let us consider the *vector space* of square $\rho$-integrable functions $\mathcal{L}^2(\rho) = \{u \in \mathbb{R}^\mathcal{S} : \int_\mathcal{S} u(s)^2\rho(ds) < \infty\}$. The measure $\rho$ naturally induces an inner-product in this space, $\langle \cdot, \cdot \rangle_\rho : \mathcal{L}^2(\rho) \times \mathcal{L}^2(\rho) \to [0, \infty)$, defined as $\langle v_1, v_2 \rangle_\rho = \int_\mathcal{S} v_1(s)v_2(s)\rho(ds)$, which is the expected correlation of the input vectors $v_1, v_2 \in \mathcal{L}^2(\rho)$ under the measure $\rho$. With this, we can define the transition operator $P_\pi : \mathcal{L}^2(\rho) \to \mathcal{L}^2(\rho)$ as:

$$[P_\pi v](s) = \int_\mathcal{S} v(s')p_\pi(s'|s)\rho(ds') \,.$$

Note that, just as the original transition matrix $\mathbf{P}_\pi$ mapped vectors in $\mathbb{R}^{|\mathcal{S}|}$ to itself, the transition operator does the same in $\mathcal{L}^2(\rho)$. Hence, it is its abstract analogous and, correspondingly, we can define the abstract graph Laplacian as the square $\rho$-integrable linear operator $L = I - f(P_\pi)$ in $\text{End}(\mathcal{L}^2(\rho))$, where $I$ is the identity linear operator and $f : \text{End}(\mathcal{L}^2(\rho)) \to \text{End}(\mathcal{L}^2(\rho))$ is a function that maps $P_\pi$ to a *self-adjoint* square $\rho$-integrable linear operator $f(P_\pi)$.[9] That a linear operator is self-adjoint means, essentially, that its corresponding density with respect to $\rho$ is symmetric. Thus, this restriction is equivalent to ask that $\mathbf{L}$ is a symmetric matrix in the finite-dimensional case. Hence, when $p_\pi(s'|s) \neq p_\pi(s|s')$, we can define $f$ in such a way that the density of $f(P_\pi)$ is $\frac{1}{2}(p_\pi(s|s') + p_\pi(s'|s))$ (see Equation 4 by Wu et al., 2019).

**Extension of Lemma 2.** After the introduced redefinitions, ALLO takes the exact same functional form as in Equation (6):

$$\max_{\boldsymbol{\beta}} \min_{u_1, \cdots, u_d \in \mathcal{L}^2(\rho)} \quad \sum_{i=1}^d \langle u_i, Lu_i \rangle_\rho + \sum_{j=1}^d \sum_{k=1}^j \beta_{jk}\big(\langle u_j, [\![u_k]\!]\rangle_\rho - \delta_{jk}\big) + \cdots \tag{20}$$

$$\cdots + b\sum_{j=1}^d \sum_{k=1}^j \big(\langle u_j, [\![u_k]\!]\rangle_\rho - \delta_{jk}\big)^2 \,.$$

In the abstract setting, and in general, we can define the differential $DF$ of a linear operator $F$ as:

$$DF(u)[v] = \frac{d}{dt}L(u + tv)\big|_{t=0} = \lim_{t \to \infty} \frac{L(u+tv) - Lu}{t} \,.$$

Now, let us fix the dual variables $\boldsymbol{\beta}$, an index $1 \leq i \leq d$, and the functions $u_j$ with $j \neq i$, We denote our ALLO objective with these fixed values by $F_i = F_i(\boldsymbol{\beta}, (u_j)_{j \neq i}) : \mathcal{L}^2(\rho) \to \mathcal{L}^2(\rho)$. Then, the change of this operator at the function $u_i$ in a direction $v_i$, with step size $t$, where the stop gradient operators affect $t$ now,[10] is:

$$F_i(u_i + tv_i) = \left[ \sum_{j \neq i} \langle u_j, Lu_j \rangle_\rho + \sum_{j=1}^{i-1} \sum_{k=1}^j \beta_{jk}\big(\langle u_j, [\![u_k]\!]\rangle_\rho - \delta_{jk}\big) + \cdots \right.$$

$$\cdots + \sum_{j=i+1}^d \sum_{\substack{k=1 \\ k \neq i}}^j \beta_{jk}\big(\langle u_j, [\![u_k]\!]\rangle_\rho - \delta_{jk}\big) + b\sum_{j=1}^{i-1} \sum_{k=1}^j \big(\langle u_j, [\![u_k]\!]\rangle_\rho - \delta_{jk}\big)^2 + \cdots$$

$$\cdots + b\sum_{j=i+1}^d \sum_{\substack{k=1 \\ k \neq i}}^j \big(\langle u_j, [\![u_k]\!]\rangle_\rho - \delta_{jk}\big)^2 \right] + \langle u_i + tv_i, L(u_i + tv_i) \rangle_\rho + \cdots$$

---

[9]$\text{End}(\mathcal{L}^2(\rho))$ denotes the space of endomorphisms of $\mathcal{L}^2(\rho)$, i.e., the space of linear operators between $\mathcal{L}^2(\rho)$ and itself.

[10]This means that $[\![u_i + tv_i]\!] = [\![u_i]\!] = u_i$.

$$\cdots + \beta_{ii}\big(\langle(u_i + tv_i), [\![u_i + tv_i]\!]\rangle_\rho - \delta_{ii}\big) + b\big(\langle(u_i + tv_i), [\![u_i + tv_i]\!]\rangle_\rho - \delta_{ii}\big)^2 + \cdots$$

$$\cdots + \sum_{k=1}^{i-1} \beta_{ik}\big(\langle(u_i + tv_i), [\![u_k]\!]\rangle_\rho - \delta_{ik}\big) + \sum_{k=1}^{i-1} b\big(\langle(u_i + tv_i), [\![u_k]\!]\rangle_\rho - \delta_{ik}\big)^2 + \cdots$$

$$\cdots + \sum_{j=i+1}^{d} \beta_{ji}\big(\langle u_j, [\![u_i + tv_i]\!]\rangle_\rho - \delta_{ji}\big) + \sum_{j=i+1}^{d} b\big(\langle u_j, [\![u_i + tv_i]\!]\rangle_\rho - \delta_{ji}\big)^2$$

$$= F_i(u_i) + t\Bigg[2\langle u_i, Lv_i\rangle_\rho + \beta_{ii}\langle u_i, v_i\rangle_\rho + 2b\langle u_i, v_i\rangle_\rho\big(\langle(u_i, [\![u_i]\!]\rangle_\rho - \delta_{ii}\big) + \cdots$$

$$\cdots + \sum_{k=1}^{i-1} \beta_{ik}\langle u_k, v_i\rangle_\rho + 2\sum_{k=1}^{i-1} b\langle u_k, v_i\rangle_\rho\big(\langle(u_i, [\![u_k]\!]\rangle_\rho - \delta_{ik}\big)\Bigg] + \mathcal{O}(t^2).$$

Hence, we can conclude that its differential, $DF_i$, is given by:

$$DF_i(u_i)[v_i] = \lim_{t\to\infty} \frac{F_i(u_i + tv_i) - F_i(u - i)}{t}$$

$$= 2\langle u_i, Lv_i\rangle_\rho + \beta_{ii}\langle u_i, v_i\rangle_\rho + 2b\langle u_i, v_i\rangle_\rho\big(\langle(u_i, [\![u_i]\!]\rangle_\rho - \delta_{ii}\big) + \cdots$$

$$\cdots + \sum_{k=1}^{i-1} \beta_{ik}\langle u_k, v_i\rangle_\rho + 2\sum_{k=1}^{i-1} b\langle u_k, v_i\rangle_\rho\big(\langle(u_i, [\![u_k]\!]\rangle_\rho - \delta_{ik}\big)$$

$$= \Bigg\langle v_i \ , \ 2Lu_i + \beta_{ii}u_i + 2b\big(\langle u_i, [\![u_i]\!]\rangle_\rho - \delta_{ii}\big)u_i + \sum_{k=1}^{i-1} \beta_{ik}u_k + \cdots$$

$$\cdots + 2b\sum_{k=1}^{i-1}\big(\langle u_i, [\![u_k]\!]\rangle_\rho - \delta_{ik}\big)u_k\Bigg\rangle_\rho$$

$$= \Bigg\langle v_i \ , \ 2Lu_i + \sum_{k=1}^{i} \beta_{ik}u_k + 2b\sum_{k=1}^{i}\big(\langle u_i, u_k\rangle_\rho - \delta_{ik}\big)u_k\Bigg\rangle_\rho.$$

Now, we have that the gradient in an arbitrary inner-product vector space is defined as the vector $\nabla F(u)$ such that $DF(u)[v] = \langle v, \nabla F(u)\rangle_\rho$. That is, the gradient of ALLO with respect to $u_i$, given the stop gradient operators, denoted as $g_i(\boldsymbol{\beta}, (u_i)_{i=1}^d)$, is:

$$g_i(\boldsymbol{\beta}, (u_i)_{i=1}^d) := \nabla F_i(u_i) = 2Lu_i + \sum_{k=1}^{i} \beta_{ik}u_k + 2b\sum_{k=1}^{i}\big(\langle u_i, u_k\rangle_\rho - \delta_{ik}\big)u_k.$$

We can note that this expression is exactly equivalent to the one obtained in the finite dimensional case in Equation (7). Similarly, one can derive an analogous to Equation (8), and, since the proof of Lemma 2 only depends on the properties of the inner product, and not a specific inner product, we can conclude that Lemma 2 applies to the general abstract setting as well. In particular, we have that the set of permutations of $d$ eigenvectors of the Laplacian is equal to the equilibria of the gradient ascent-descent dynamics for ALLO.[11]

**Extension of Theorem 1.**  By analogy with the previous derivation, we can obtain the Hessian of $F_i$ with respect to $u_j$, considering the stop gradients (i.e., the gradient of $g_i$), by replacing matrices

---

[11]Note that our derivation is exactly the same one would follow to calculate the gradient in the finite dimensional case if one only uses the definitions of differential and gradient. Under this light, the fact that we are considering abstract spaces instead of finite dimensional ones seems irrelevant for Lemma 2.

with their linear operator counterparts in Equations (12)-(15). Then, since the Laplacian is a self-adjoint square $\rho$-integrable linear operator, typically called Hilbert-Schmidt integral operator, it is also a compact operator. Thus, any vector can be expressed as a countable sum of eigenvectors, i.e., there are potentially infinite, but at most countable eigenvectors, and so the steps from Equation (17) to Equation (19) still hold, due to the *spectral theory of compact operators*.

## B  ENVIRONMENTS

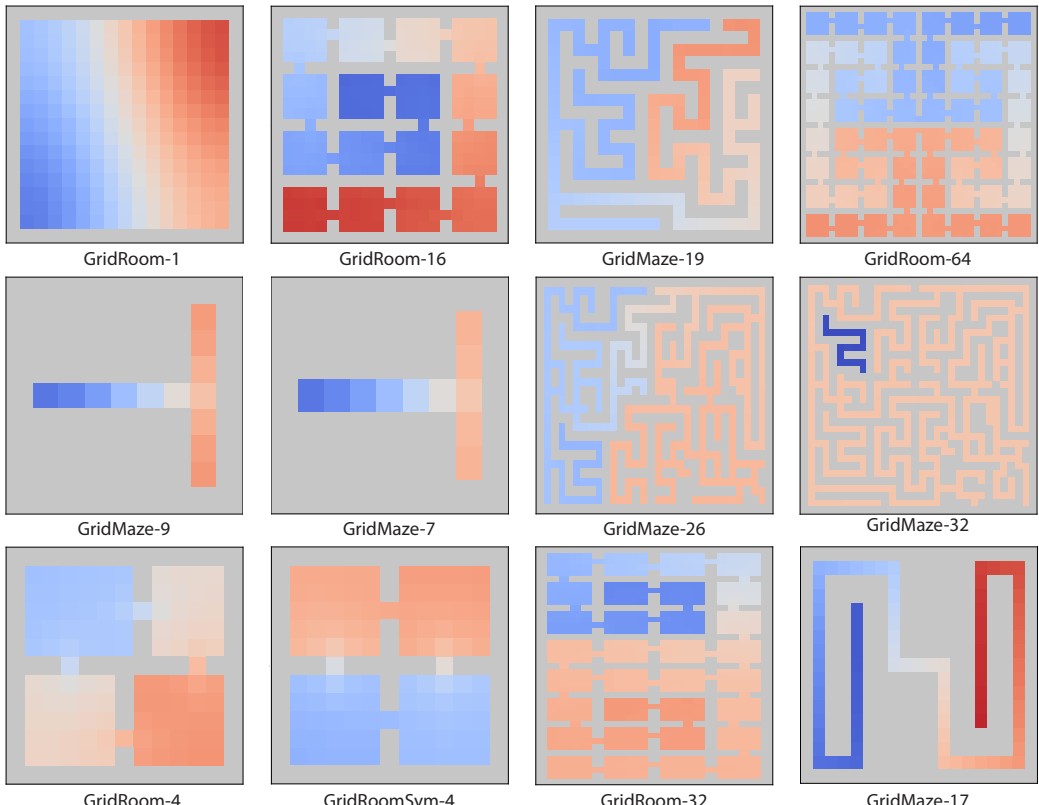

Figure 6: Grid environments where the Laplacian representation is learned with both GGDO and ALLO. Color corresponds to the second smallest eigenvector of the Laplacian learned by ALLO.

## C  LEARNED EIGENVECTORS

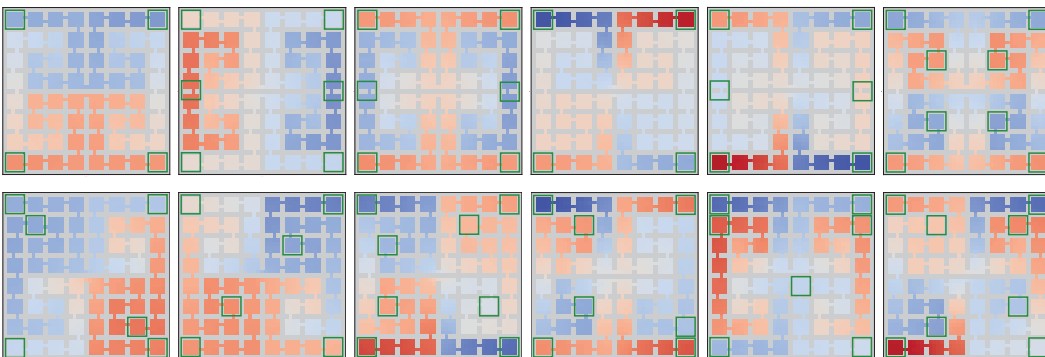

Figure 7: Second to seventh Laplacian eigenvectors in increasing order for the environment `GridRoom-64`. The top row corresponds to ALLO, while the bottom one to GGDO. The green bounding boxes indicate the location of a local minimum or maximum.

Figure 7 shows the difference between the smallest eigenvectors of the Laplacian (those learned via ALLO) and rotations of them (those learned via GGDO). One can note how, in general, the rotations present two characteristics: they have more local minima and local maxima, and their location is distributed irregularly in the state space. The first characteristic is relevant considering that the eigenvectors are used as reward functions to discover options, i.e., temporally-extended actions, that

have as sink states these critical states. The more sink states, the shorter the trajectories associated with an induced option, and, as a result, **the less exploration**.

Similarly, the irregularity is a manifestation of eigenvectors of different temporal scales being combined, which affects the temporal distance induced by the Laplacian representation. In particular, as studied by Wang et al. (2023), if each eigenvector is normalized by their corresponding eigenvalue, the representation distance corresponds exactly to the (square root of the) temporal distance. However, the eigenvalues, which are a measure of the temporal scale (see Jinnai et al., 2019), are only defined for the original eigenvectors. Hence, the distance induced by the permutations cannot be normalized and, as a result, **it is not as suitable for reward shaping**.

## D  LEARNING WITH RANDOM PERMUTATIONS

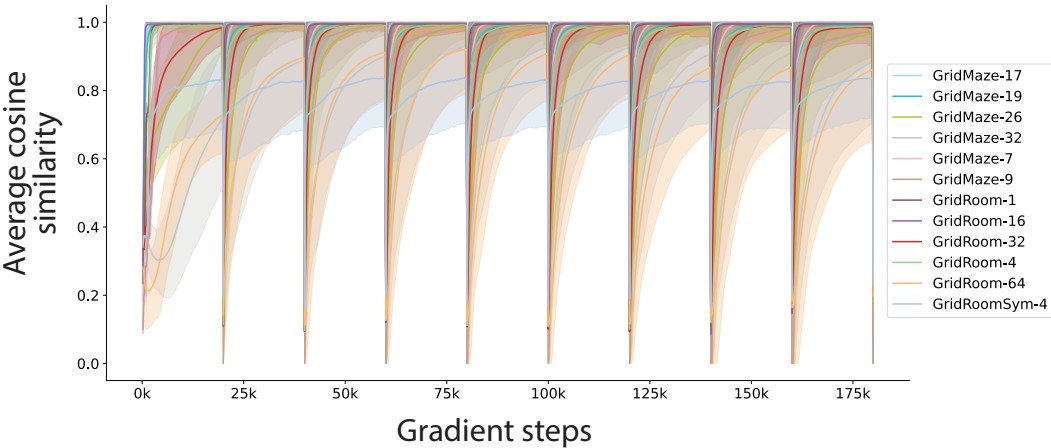

Figure 8: Average cosine similarity between the true Laplacian representation and ALLO for $\alpha_{\text{barrier}} = 0.01$, averaged over 60 seeds. Each 20,000 gradient steps the coordinates of the Laplacian representation are randomly permuted. The shaded region corresponds to a 95% confidence interval.

As an additional sanity check for Theorem 1, Figure 8 shows the evolution of the cosine similarity for the 12 grid environments considered under random permutations. Specifically, each 20,000 gradient steps a random permutation is sampled and then the output coordinates of the approximator $\phi_\theta$ are permuted accordingly. If any arbitrary permutation (or rotation) of the Laplacian eigenvectors were stable, the cosine similarity would decrease after the permutation and it would not increase again. On the contrary, we can observe how in each case learning is as successful as in the initial cycle without permutations.

## E  AVERAGE COSINE SIMILARITY COMPARISON

In addition to the results considered in Figure 4, Figure 9b shows the cosine similarity for *each* of the components of the Laplacian representation. It is worth noting that in some environments (e.g., `GridRoom-32` and `GridMaze-32`) GGDO struggles with the first 2 eigenvectors, which are the most relevant to induce options for exploration.

As a supplementary robustness test, we changed the state representation used as input to the neural network. In particular, we used a pixel representation where each tile in the subplots of Figure 6 corresponds to a single pixel. In this way, the inputs ranged from a size of $7 \times 9 \times 3$ to a size of $41 \times 41 \times 3$. Correspondingly, we added two convolutional layers with no pooling, stride of 2, and kernel size of 3 to the previous fully connected network.

Figure 10 contains analogous cosine similarity comparisons to those observed in Figure 9 for the $(x, y)$ state representation. There are three main differences with the previous results. First, the average cosine similarity difference (Figure 10a) is now only significant in 7 out of 12 environments, instead of 9 of 12 (see Tables 1 and 2), but the difference is higher for several environments (e.g., `GridRoom-32` and `GridRoom-1`). Second, ALLO only finds the true Laplacian representation

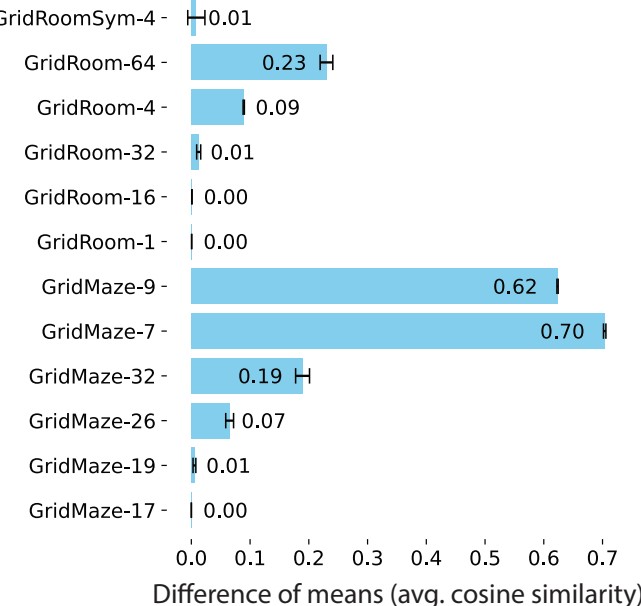

(a) Average cosine similarity across the $d$ components.

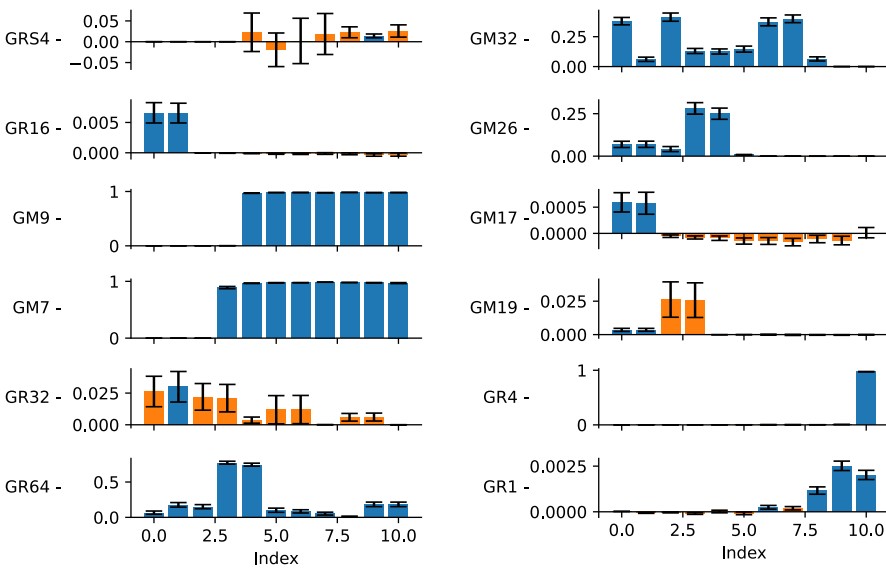

(b) Cosine similarity for each of the components. `GR` and `GM` stand for `GridRoom` and `GridMaze`. black bars correspond to p-values below $0.01$.

Figure 9: Cosine similarity difference between ALLO and GGDO when using the $(x, y)$ **state representation**. Error bars show the standard deviation of the differences, defined as $\sqrt{\sigma_{\text{ALLO}}^2/n_{\text{ALLO}} + \sigma_{\text{GGDO}}^2/n_{\text{GGDO}}}$, where $\sigma$ denotes the sample standard deviation, and $n$ the number of seeds ($= 60$ in all cases).

in 6 out of 12 environments, as opposed to 11 out of 12. From those 6 where perfect learning is not achieved, 4 have a cosine similarity above 0.9, one corresponds to the same environment where the Laplacian representation was not perfectly learned before, and there is one for which the similarity is only 0.61. Focusing on these results, we observed that ALLO was able to recover the Laplacian representation for some seeds, but for others the output of the neural network collapsed to a constant.

This suggests that maintaining the same hyperparameters with the new state representation and convolutional layers could have resulted in divergence problems unrelated to ALLO. Lastly, we can observe (see Figure 10b) that GGDO fails more frequently, in contrast with the $(x, y)$ representation case, in finding the smallest non-constant eigenvector (e.g., in `GridRoom-16` and `GridRoom-1`), supporting the hypothesis that GGDO is not scalable to more complex settings, as opposed to ALLO, and preventing the use of the Laplacian for exploration.

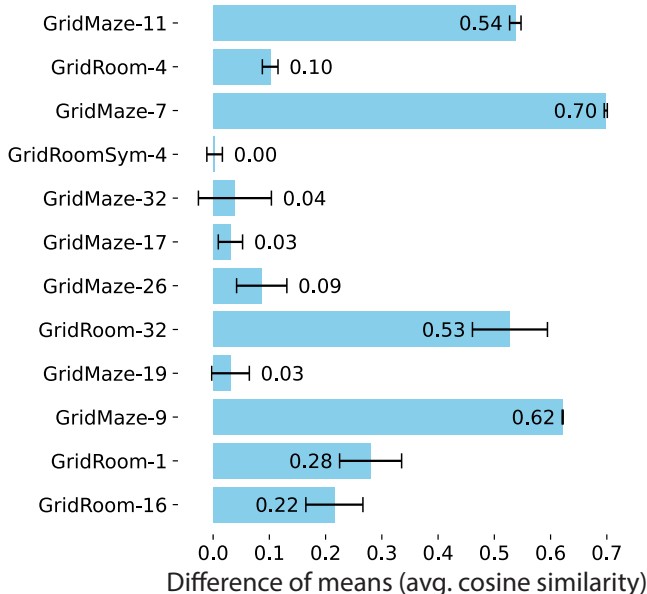

(a) Average cosine similarity across the $d$ components.

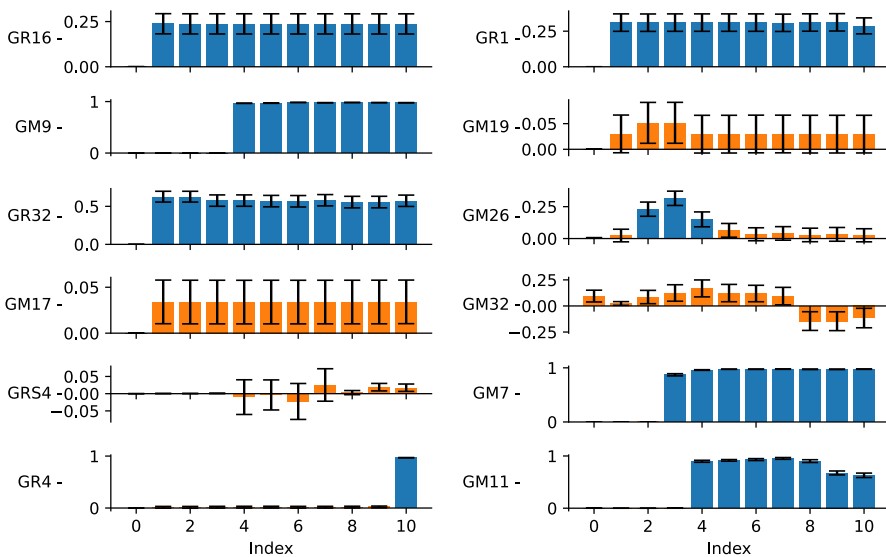

(b) Cosine similarity for each of the components. `GR` and `GM` stand for `GridRoom` and `GridMaze`. black bars correspond to p-values below 0.01.

Figure 10: Cosine similarity difference between ALLO and GGDO when using the **pixel state representation**. Error bars show the standard deviation of the differences, defined as $\sqrt{\sigma_{\text{ALLO}}^2/n_{\text{ALLO}} + \sigma_{\text{GGDO}}^2/n_{\text{GGDO}}}$, where $\sigma$ denotes the sample standard deviation, and $n$ the number of seeds ($\in [50, 60]$ in all cases).

| Env | ALLO | GGDO | t-statistic | p-value |
|---|---|---|---|---|
| GridMaze-17 | 0.9994 (0.0002) | 0.9993 (0.0003) | 0.641 | 0.262 |
| GridMaze-19 | 0.9989 (0.0006) | 0.9936 (0.0185) | 2.218 | 0.015 |
| GridMaze-26 | **0.9984 (0.0007)** | 0.9331 (0.0517) | 9.770 | 0.000 |
| GridMaze-32 | **0.9908 (0.0161)** | 0.8014 (0.0901) | 16.018 | 0.000 |
| GridMaze-7 | **0.9996 (0.0002)** | 0.2959 (0.0159) | 343.724 | 0.000 |
| GridMaze-9 | **0.9989 (0.0007)** | 0.3755 (0.0081) | 596.775 | 0.000 |
| GridRoom-1 | **0.9912 (0.0003)** | 0.9906 (0.0003) | 9.691 | 0.000 |
| GridRoom-16 | **0.9990 (0.0004)** | 0.9980 (0.0023) | 3.297 | 0.001 |
| GridRoom-32 | **0.9982 (0.0010)** | 0.9857 (0.0266) | 3.647 | 0.000 |
| GridRoom-4 | **0.9965 (0.0052)** | 0.9073 (0.0063) | 84.136 | 0.000 |
| GridRoom-64 | **0.9917 (0.0059)** | 0.7617 (0.0834) | 21.326 | 0.000 |
| GridRoomSym-4 | 0.8411 (0.0742) | 0.8326 (0.0855) | 0.581 | 0.281 |

Table 1: Average cosine similarities for ALLO and GGDO when using the $(x, y)$ **state representation**. The sample average is calculated using 60 seeds. The standard deviation is shown in parenthesis and the maximal average is shown in boldface when the difference is statistically significant, meaning the associated p-value is smaller than 0.01.

| Env | ALLO | GGDO | t-statistic | p-value |
|---|---|---|---|---|
| GridRoom-16 | **0.9992 (0.0003)** | 0.7836 (0.3897) | 4.251 | 0.000 |
| GridRoom-1 | **0.9914 (0.0002)** | 0.7114 (0.4199) | 5.080 | 0.000 |
| GridMaze-9 | **0.9988 (0.0013)** | 0.3773 (0.0064) | 720.882 | 0.000 |
| GridMaze-19 | 0.9789 (0.1354) | 0.9480 (0.2032) | 0.926 | 0.178 |
| GridRoom-32 | **0.9073 (0.2591)** | 0.3795 (0.4183) | 7.917 | 0.000 |
| GridMaze-26 | 0.9330 (0.2214) | 0.8466 (0.2358) | 1.935 | 0.028 |
| GridMaze-17 | 0.9995 (0.0001) | 0.9687 (0.1658) | 1.429 | 0.079 |
| GridMaze-32 | 0.6198 (0.3056) | 0.5810 (0.3745) | 0.598 | 0.276 |
| GridRoomSym-4 | 0.8434 (0.0743) | 0.8407 (0.0686) | 0.197 | 0.422 |
| GridMaze-7 | **0.9997 (0.0002)** | 0.3015 (0.0211) | 253.863 | 0.000 |
| GridRoom-4 | **0.9965 (0.0076)** | 0.8950 (0.1065) | 7.297 | 0.000 |
| GridMaze-11 | **0.9349 (0.0496)** | 0.3975 (0.0592) | 52.931 | 0.000 |

Table 2: Average cosine similarities for ALLO and GGDO when using the **pixel state representation**. The sample average is calculated using 60 seeds. The standard deviation is shown in parenthesis and the maximal average is shown in boldface when the difference is statistically significant, meaning the associated p-value is smaller than 0.01.

# F   AVERAGE EIGENVALUES

| Env | True | ALLO | GGDO | Env | True | ALLO | GGDO |
|---|---|---|---|---|---|---|---|
| GridRoomSym-4 | 0.0000 | 0.0000 (0.0000) | 0.0000 (0.0000) | GridMaze-32 | 0.0000 | 0.0000 (0.0000) | 0.0001 (0.0000) |
| | 0.0540 | 0.0429 (0.0003) | 0.0395 (0.0003) | | 0.0000 | 0.0007 (0.0001) | 0.0004 (0.0001) |
| | 0.0540 | 0.0434 (0.0003) | 0.0387 (0.0003) | | 0.0004 | 0.0008 (0.0000) | 0.0006 (0.0000) |
| | 0.1068 | 0.0863 (0.0005) | 0.0703 (0.0005) | | 0.0013 | 0.0015 (0.0000) | 0.0010 (0.0000) |
| | 0.4502 | 0.4034 (0.0007) | 0.0531 (0.0007) | | 0.0037 | 0.0035 (0.0000) | 0.0033 (0.0000) |
| | 0.4507 | 0.4045 (0.0007) | 0.0529 (0.0007) | | 0.0043 | 0.0040 (0.0000) | 0.0038 (0.0000) |
| | 0.4507 | 0.4054 (0.0007) | 0.0524 (0.0007) | | 0.0058 | 0.0053 (0.0001) | 0.0050 (0.0001) |
| | 0.4512 | 0.4066 (0.0008) | 0.0520 (0.0008) | | 0.0063 | 0.0056 (0.0001) | 0.0053 (0.0001) |
| | 0.4915 | 0.4479 (0.0008) | 0.0156 (0.0008) | | 0.0075 | 0.0067 (0.0001) | 0.0059 (0.0001) |
| | 0.4951 | 0.4513 (0.0007) | 0.0121 (0.0007) | | 0.0099 | 0.0085 (0.0001) | 0.0080 (0.0001) |
| | 0.4951 | 0.4526 (0.0008) | 0.0108 (0.0008) | | 0.0148 | 0.0126 (0.0001) | 0.0118 (0.0001) |
| GridRoom-16 | 0.0000 | 0.0000 (0.0000) | 0.0000 (0.0000) | GridMaze-26 | 0.0000 | 0.0000 (0.0000) | 0.0001 (0.0000) |
| | 0.0016 | 0.0017 (0.0000) | 0.0016 (0.0000) | | 0.0005 | 0.0008 (0.0000) | 0.0007 (0.0000) |
| | 0.0063 | 0.0055 (0.0000) | 0.0051 (0.0000) | | 0.0034 | 0.0032 (0.0000) | 0.0030 (0.0000) |
| | 0.0139 | 0.0116 (0.0001) | 0.0111 (0.0001) | | 0.0039 | 0.0037 (0.0001) | 0.0035 (0.0001) |
| | 0.0242 | 0.0199 (0.0002) | 0.0189 (0.0002) | | 0.0048 | 0.0044 (0.0001) | 0.0039 (0.0001) |
| | 0.0367 | 0.0301 (0.0002) | 0.0279 (0.0002) | | 0.0058 | 0.0051 (0.0001) | 0.0048 (0.0001) |
| | 0.0511 | 0.0420 (0.0002) | 0.0382 (0.0002) | | 0.0094 | 0.0080 (0.0001) | 0.0078 (0.0001) |
| | 0.0663 | 0.0546 (0.0003) | 0.0482 (0.0003) | | 0.0129 | 0.0109 (0.0001) | 0.0105 (0.0001) |
| | 0.0832 | 0.0688 (0.0005) | 0.0591 (0.0005) | | 0.0156 | 0.0131 (0.0001) | 0.0130 (0.0001) |
| | 0.1007 | 0.0836 (0.0006) | 0.0691 (0.0006) | | 0.0195 | 0.0164 (0.0002) | 0.0154 (0.0002) |
| | 0.1161 | 0.0969 (0.0005) | 0.0774 (0.0005) | | 0.0255 | 0.0213 (0.0002) | 0.0201 (0.0002) |
| GridMaze-9 | 0.0000 | 0.0001 (0.0000) | 0.0000 (0.0000) | GridMaze-17 | 0.0000 | 0.0000 (0.0000) | 0.0000 (0.0000) |
| | 0.1582 | 0.1154 (0.0006) | 0.0962 (0.0006) | | 0.0029 | 0.0027 (0.0000) | 0.0026 (0.0000) |
| | 0.3083 | 0.2354 (0.0011) | 0.1159 (0.0011) | | 0.0116 | 0.0097 (0.0001) | 0.0094 (0.0001) |
| | 0.4899 | 0.3961 (0.0015) | 0.0181 (0.0015) | | 0.0257 | 0.0212 (0.0001) | 0.0199 (0.0001) |
| | 0.6613 | 0.5674 (0.0020) | 0.0000 (0.0020) | | 0.0448 | 0.0368 (0.0002) | 0.0331 (0.0002) |
| | 0.7529 | 0.6692 (0.0024) | 0.0000 (0.0024) | | 0.0682 | 0.0562 (0.0003) | 0.0490 (0.0003) |
| | 0.7777 | 0.6986 (0.0023) | 0.0000 (0.0023) | | 0.0952 | 0.0790 (0.0004) | 0.0647 (0.0004) |
| | 0.8266 | 0.7585 (0.0027) | 0.0000 (0.0027) | | 0.1251 | 0.1045 (0.0005) | 0.0809 (0.0005) |
| | 0.8613 | 0.8038 (0.0028) | 0.0000 (0.0028) | | 0.1572 | 0.1327 (0.0005) | 0.0950 (0.0005) |
| | 0.8768 | 0.8251 (0.0029) | 0.0000 (0.0029) | | 0.1907 | 0.1624 (0.0006) | 0.1065 (0.0006) |
| | 0.8796 | 0.8292 (0.0029) | 0.0000 (0.0029) | | 0.2249 | 0.1937 (0.0007) | 0.1163 (0.0007) |
| GridMaze-7 | 0.0000 | 0.0001 (0.0000) | 0.0000 (0.0000) | GridMaze-19 | 0.0000 | 0.0000 (0.0000) | 0.0000 (0.0000) |
| | 0.1833 | 0.1325 (0.0006) | 0.1034 (0.0006) | | 0.0016 | 0.0016 (0.0000) | 0.0014 (0.0000) |
| | 0.4622 | 0.3645 (0.0011) | 0.0437 (0.0011) | | 0.0058 | 0.0051 (0.0000) | 0.0048 (0.0000) |
| | 0.5208 | 0.4186 (0.0010) | 0.0000 (0.0010) | | 0.0068 | 0.0059 (0.0001) | 0.0056 (0.0001) |
| | 0.7148 | 0.6158 (0.0012) | 0.0000 (0.0012) | | 0.0140 | 0.0117 (0.0001) | 0.0111 (0.0001) |
| | 0.7958 | 0.7084 (0.0014) | 0.0000 (0.0014) | | 0.0232 | 0.0192 (0.0002) | 0.0177 (0.0002) |
| | 0.8392 | 0.7621 (0.0014) | 0.0000 (0.0014) | | 0.0365 | 0.0300 (0.0002) | 0.0283 (0.0002) |
| | 0.8549 | 0.7826 (0.0017) | 0.0000 (0.0017) | | 0.0403 | 0.0332 (0.0003) | 0.0306 (0.0003) |
| | 0.8739 | 0.8076 (0.0015) | 0.0000 (0.0015) | | 0.0516 | 0.0425 (0.0004) | 0.0390 (0.0004) |
| | 0.8934 | 0.8347 (0.0016) | 0.0000 (0.0016) | | 0.0559 | 0.0461 (0.0003) | 0.0416 (0.0003) |
| | 0.9091 | 0.8572 (0.0018) | 0.0000 (0.0018) | | 0.0821 | 0.0679 (0.0004) | 0.0567 (0.0004) |
| GridRoom-32 | 0.0000 | 0.0000 (0.0000) | 0.0000 (0.0000) | GridRoom-4 | 0.0000 | 0.0000 (0.0000) | 0.0000 (0.0000) |
| | 0.0008 | 0.0010 (0.0000) | 0.0008 (0.0000) | | 0.0490 | 0.0392 (0.0003) | 0.0357 (0.0003) |
| | 0.0018 | 0.0019 (0.0000) | 0.0017 (0.0000) | | 0.0576 | 0.0462 (0.0003) | 0.0416 (0.0003) |
| | 0.0039 | 0.0036 (0.0000) | 0.0031 (0.0000) | | 0.1122 | 0.0908 (0.0005) | 0.0732 (0.0005) |
| | 0.0065 | 0.0057 (0.0001) | 0.0053 (0.0001) | | 0.3905 | 0.3448 (0.0014) | 0.0923 (0.0014) |
| | 0.0135 | 0.0114 (0.0001) | 0.0105 (0.0001) | | 0.4420 | 0.3963 (0.0009) | 0.0592 (0.0009) |
| | 0.0161 | 0.0135 (0.0001) | 0.0124 (0.0001) | | 0.4531 | 0.4080 (0.0010) | 0.0517 (0.0010) |
| | 0.0200 | 0.0167 (0.0001) | 0.0151 (0.0001) | | 0.4585 | 0.4139 (0.0010) | 0.0451 (0.0010) |
| | 0.0270 | 0.0223 (0.0002) | 0.0206 (0.0002) | | 0.4787 | 0.4348 (0.0008) | 0.0283 (0.0008) |
| | 0.0284 | 0.0236 (0.0002) | 0.0212 (0.0002) | | 0.4917 | 0.4489 (0.0010) | 0.0152 (0.0010) |
| | 0.0364 | 0.0301 (0.0002) | 0.0265 (0.0002) | | 0.5209 | 0.4802 (0.0008) | 0.0000 (0.0008) |
| GridRoom-64 | 0.0000 | 0.0000 (0.0000) | 0.0001 (0.0000) | GridRoom-1 | 0.0000 | 0.0000 (0.0000) | 0.0000 (0.0000) |
| | 0.0004 | 0.0007 (0.0000) | 0.0006 (0.0000) | | 0.0895 | 0.0728 (0.0003) | 0.0610 (0.0003) |
| | 0.0010 | 0.0012 (0.0000) | 0.0009 (0.0000) | | 0.0895 | 0.0735 (0.0004) | 0.0629 (0.0004) |
| | 0.0016 | 0.0017 (0.0000) | 0.0017 (0.0000) | | 0.1643 | 0.1372 (0.0005) | 0.0986 (0.0005) |
| | 0.0020 | 0.0021 (0.0000) | 0.0015 (0.0000) | | 0.2801 | 0.2412 (0.0006) | 0.1175 (0.0006) |
| | 0.0021 | 0.0022 (0.0000) | 0.0018 (0.0000) | | 0.2801 | 0.2425 (0.0005) | 0.1183 (0.0005) |
| | 0.0035 | 0.0033 (0.0000) | 0.0031 (0.0000) | | 0.3277 | 0.2868 (0.0007) | 0.1128 (0.0007) |
| | 0.0041 | 0.0038 (0.0000) | 0.0034 (0.0000) | | 0.3277 | 0.2884 (0.0007) | 0.1134 (0.0007) |
| | 0.0063 | 0.0055 (0.0000) | 0.0051 (0.0000) | | 0.4376 | 0.3980 (0.0009) | 0.0622 (0.0009) |
| | 0.0090 | 0.0078 (0.0001) | 0.0072 (0.0001) | | 0.4622 | 0.4229 (0.0008) | 0.0432 (0.0008) |
| | 0.0097 | 0.0084 (0.0001) | 0.0078 (0.0001) | | 0.4622 | 0.4244 (0.0008) | 0.0419 (0.0008) |

Table 3: Average eigenvalues for ALLO and GGDO. For each environment, the true eigenvalues are shown in decreasing order, from the 2nd one up to the 11th. The sample averages are calculated using 60 seeds and in parenthesis are the standard deviations.

