# OpenReview forum: "Proper Laplacian Representation Learning"
_ICLR.cc/2024/Conference — ICLR 2024 poster_

### Official Review · Reviewer_tVNo · 2023-10-13

**Soundness:** 4 excellent
**Presentation:** 4 excellent
**Contribution:** 3 good
**Rating:** 6
**Confidence:** 4

**Summary:**

This paper proposes a new method for approximating the graph Laplacian (and namely its eigenvectors) over a discrete state space. This paper builds off of the "graph drawing objective" (and "generalized graph drawing objective"), with the goal to eliminate hyperparameters in its optimization that have been shown to sensitively effect the output approximation of the graph Laplacian's eigenvectors. It achieves this goal with a reformulated minimax objective, and provides both theory and experiments to justify this novel objective.

**Strengths:**

- This is a very mathematically clear and concise paper. The objective of the paper is clear and well-presented (re-formulate the graph drawing objective with a smaller hyperparameter space), the solution is clearly presented (convert into a max-min game, in spirit as a replacement to these hyperparameters), and the theory is both mathematically sound and interpretable to understand why the max-min game here achieves the desired objective.
- While the proposed solution appears "simple" on paper, the theory and justification behind the method is both theoretically rich and clever, combining both a nice environment for theoreticians and a direct benefit for practitioners (less hyperparameters to tune alongside the rest of the system).

**Weaknesses:**

- The primary limitation is my view is the lack of justification in the practical context. Experiments are only provided in very simple maze scenarios, and it is not demonstrated here (or in primary related work I saw) why, in practice, one would chose a Laplacian representation over a standard representation.
  - On a similar note, there lacks more thorough discussion on the stability of the now-induced minimax game. This likewise seems like something necessary in order to demonstrate practical utility of this framework, at least theoretically.

Nonetheless, I feel these weaknesses do not bar this paper from being a good publication as it is. It very clearly and concisely establishes the new method for Laplacian representation learning, and even if this framework is not currently in mainstream practical usage, it gives a solid and approachable platform for future research in improving both theory (e.g. stability of the minimax game could constitute an entirely separate paper) and practice (e.g. implementing standard RL engineering tricks to push practical performance over standard RL methods in certain scenarios).

(However, as a side note, perhaps the current title is a bit presumptuous until such further theory and experimentation has been established.)

**Questions:**

- As there are natural continuous analogs to the graph Laplacion in Euclidean spaces, I am curious how, at least in theory, this framework is extendible into continuous state and action spaces? What are the limitations in extending this theory to continuous settings?

---

> ### Author Response · Authors · 2023-11-18
>
> We thank the reviewer for pointing out how in its initial version the manuscript seemed to suggest that our derivations only applied to the tabular setting where the state space is finite. On the contrary, both theory and experiments apply to the abstract setting, meaning that the state space is a measure space that is potentially uncountable. This is more general than the case in which the state space is a continuous Euclidean space, and therefore applies to it. To make this unambiguous, we included a whole new subsection in the Appendix (Appendix A.5) that details how the theory developed in Section 4 applies to the abstract setting.
>
> We also included a final paragraph in Section 2 that makes this explicit and that connects the abstract/continuous setting with the use of neural network approximators. Something to note here is that the action space we consider is still discrete, as in all the previous Laplacian literature. However, we do not see any relationship between this assumption and our objective since, once the policy is fixed, what matters is the resulting Laplacian, whose dimensionality depends on the state space and not the action space.
>
> Now we would like to address the points raised as weaknesses in the review:
>
> Weakness 1: The primary limitation is my view is the lack of justification in the practical context. Experiments are only provided in very simple maze scenarios, and it is not demonstrated here (or in primary related work I saw) why, in practice, one would chose a Laplacian representation over a standard representation.
>
> Answer: The Laplacian representation is useful for specific applications such as the ones we mentioned in Section 1 (e.g., exploration, reward shaping, and preserving information relevant for value estimation) and it is not meant as a state representation suitable for solving any possible application. In this sense, the utility of the Laplacian representation has been established by previous literature (e.g., refer to Klissarov and Machado, 2023) and we do not consider necessary a further comparison with other types of representations. All this being said, we consider that we could have been more clear about the motivation of the Laplacian representation and for this reason we updated the third paragraph in Section 1.
>
> Moreover, the reason we limited the experiments to gridworlds is that the tabular setting is the only setting for which one can calculate exactly the eigenvectors of the Laplacian. Since we want to evaluate the correctness and generality of the proposed objective to learn the Laplacian representation, and not to motivate this representation, we concluded that the appropriate environment hyperparameters to vary were sizes and topologies. In particular, note that the number of states ranges from 11 (GridMaze-7) to 1088 (GridRoom-64), and the environments present both totally disconnected regions (e.g., GridMaze-32), highly connected regions (e.g., GridRoom-1), and symmetries (compare GridRoom-4 and GridRoomSym-4). Also note that the largest environments are larger than any of the tabular environments considered in the literature we reviewed (specifically, see Wu, 2019; Wang, 2021; Wang, 2023).
>
> All that being said, your invitation to consider additional experiments made us reconsider an additional environment hyperparameter that was explored by previous works. Specifically, our initial submission only considered as a representation of the state the (x, y) location of the agent in the grid. Now we include in the Appendix similar results for the pixel-based representation of the state, where each tile in the grid corresponds to a pixel (see Figure 10 and Table 2). In particular, our results continue to be promising and they continue to outperform the existing solutions, further strengthening the claims in our paper.
>
> Weakness 2: On a similar note, there lacks more thorough discussion on the stability of the now-induced minimax game. This likewise seems like something necessary in order to demonstrate practical utility of this framework, at least theoretically.
>
> Answer: We are unsure about what additional discussion regarding the stability of the induced minimax game is appropriate. We are open to any suggestion that you consider relevant. So far, we proved with Theorem 1, and the new extension to the abstract setting in the Appendix, that the Laplacian representation is the unique stable equilibrium point of the induced minimax game, meaning that we have theoretical guarantees (under no function approximation and Monte Carlo estimation of gradients) that the minimax game is stable. In addition, now you can observe in Appendix D how the only permutation that is stable is the one corresponding to the Laplacian representation.

---

> > ### Comment · Reviewer_tVNo · 2023-11-22
> >
> > Thank you for the comprehensive response and thorough updates to the paper, both have helped me understand this area of work more. As stated before, my listed weaknesses are "weak" weaknesses--I don't think these bar this paper from me recommending this paper for publication, but they also represent directions where I would have expected to see more in order to receive a higher score in the reinforcement learning section of ICLR. The additional experiments certainly validate and evaluate your proposed method as a Laplacian representation learner, but for an 8 I would have expected some experiments on more modern RL environments (I understand the motivation of gridworld as an environment for true and verifiable Laplacian eigenvalue computation, but for me a higher scoring paper would have evidence of its usability for the average ICLR-viewing RL researcher). And I apologize for the confusion, I realize "stability" is a compounded term in game theory and I am certainly not an expert in game theory. My usage of "stability" was stability of the optimization, e.g. expected condition number of the objective Hessian at the critical point. Again, I do think the provided analysis as-is is good enough for me to recommend for publication, but for a higher score I would have liked to see more detailed analysis on the stability of the expected optimization, in the sense of how often someone using this algorithm can get to the stable equilibria, to better understand the practical trade-off of casting the optimization as a minimax game (while the phrase "no free lunch" is often overused, there are many times in my experience where casting an optimization as a minimax game has benefits, but also downsides in practical training stability).

---

### Official Review · Reviewer_DES2 · 2023-10-31

**Soundness:** 3 good
**Presentation:** 3 good
**Contribution:** 2 fair
**Rating:** 5
**Confidence:** 3

**Summary:**

The paper develops three methods for smoothing in state-space models (SSMs). The idea is to assume SSMs that are non-linear and avoid other assumptions like Gaussianity when using variational inference. The drivin gidea is to preserve the temporal structure in the variational proposal. This seems to lead to what is called exponential family dynamical systems, that it a double-looped (forward and backward) chain of markovian conditionals.

**Strengths:**

Having carefully checked the exponential family derivations, the parameterization, as well as the derived ELBOs, I feel that likely they are correct and well-founded on previous related work. The use of exponential families in this context, and particularly to build the factorization into markovian conditionals is definitely a strenght. The work itself is clear and concise on the details, also mentioning limitations and reasoning on why certain decisions are taken.

**Weaknesses:**

To me the paper has two main weaknesses:

[w1] — the paper is in general concise and thorough, but written in a way that the smoothing idea is kind of lost. Particularly, technical details jump in for solving issues of previous technical details (derivations begin at the beginning of pp. 2 and finish at the end of pp. 7). In that way, the paper loses quite a lot of space, and story on the general smoothing idea that authors want to solve (and in which way they want to solve it).

[w2] — the second concern to me is the limited results. Having derived long technical details, the manuscript should at least provide results proportional to the technical development. In my opinion, the evaluation of the model is somehow short (learning of two synthetic systems (pendulum and chaotic scenario) plus analysis on convergence).

**Questions:**

Not technical questions

---

> ### Author Response · Authors · 2023-11-10
> **Review for Another Paper**
>
> This review is clearly for a different paper.  Maybe there was an error in uploading the correct review?

---

> ### Comment · Area_Chair_DypR · 2023-11-11
> **Check the author response.**
>
> Please check the author's comment and if your review submitted is the correct one.

---

### Official Review · Reviewer_Fii6 · 2023-11-01

**Soundness:** 3 good
**Presentation:** 3 good
**Contribution:** 3 good
**Rating:** 6
**Confidence:** 1

**Summary:**

In Graph Drawing Objective (GDO) and the generalized GDO, the optimization problem in Equation 1 and 3 are used to find the Laplacian representation, but this formulation allows symmetries, which lead to hyper-parameters that can lead to potential issues. The proposed method, Augmented Lagrangian Laplacian Objective (ALLO) in Equation 6, requires no hyper-parameters. In Theorem 1, they show a theoretical result on how there is a guarantee of the stability of the proposed objective function for finding Laplacian representations. The paper concludes with some experiments.

**Strengths:**

- interesting formulation and solution
- motivated problem
- having experiments

**Weaknesses:**

- some parts (e.g., Section 1 and 2) are hard to follow

**Questions:**

- How do you compare the complexity of the proposed objective function optimization problem with previous cases?





---------------------------------------------
After the rebuttal: I appreciate the authors for their response. They fully addressed my question and I decided to keep my acceptance score.

---

> ### Author Response · Authors · 2023-11-18
>
> We thank the reviewer for pointing out the absence of any comment regarding the complexity of our proposed objective. Interpreting complexity as the computational and memory complexity of an algorithm that optimizes this objective, there is an addition of ~$d^2$ parameters that have to be stored and updated. Typically, $d^2$ will be orders of magnitude smaller than the number of parameters and hyperparameters in standard neural networks (e.g., in our experiments $d^2=121$, while the number of parameters is larger than $100,000$), and so the overhead of the proposed objective is negligible. We now address this point at the end of the third paragraph in Section 3.
>
> Regarding the lack of clarity in Sections 1 and 2, we are happy to further clarify any question or confusion and make the corresponding modifications in the manuscript. However, we need to know exactly what is not clear. That being said, we edited the third paragraph in Section 1 to make more precise the properties of the Laplacian representation that make it a suitable state representation.

---

### Official Review · Reviewer_nARE · 2023-11-03

**Soundness:** 3 good
**Presentation:** 3 good
**Contribution:** 3 good
**Rating:** 6
**Confidence:** 3

**Summary:**

The authors propose a method to approximate the true eigenvalues and eigenvectors of a graph Laplacian relying on an unconstrained max-min problem solved by gradient-based optimization. This can be used to learn good representations for the states in reinforcement learning problems. In the experiments, the efficiency of the method is demonstrated together with an ablation study.

**Strengths:**

- This is an interesting and novel approach to the challenging problem of unsupervised representation learning.
- The technical part of the paper seems to be solid and reasonable, but I have not verified the theoretical results in detail.
- Both the theoretical results and the experiments support the claims.
- The paper is relatively well written.

**Weaknesses:**

I think that the proofs could have been in appendix and instead use the space for more examples, demonstrations, and clarifications.

**Questions:**

Q1. While in the paper the approach focuses on the eigenvectors of the graph Laplacian, in the experiments it is used for finding eigenfunctions. I think that further information should be provided for the actual formulation/solution of this problem.
Q2. I find Corollary 1 and the paragraph above a bit unclear. Why does an optimum of (2) and (4) imply that the constraint must be violated?
Q3. Perhaps, an experiment to test the stability of the equilibrium with respect to permutations.
Q4. Why rotated eigenvectors do not provide a good representation?

---

> ### Author Response · Authors · 2023-11-18
>
> We find valuable the questions posed by the reviewer. In particular, they let us understand what was unclear, what required more theoretical details, or what could be further clarified with additional plots. That being said, we would like to emphasize that the paper is theoretical in nature. Correspondingly, we think the proof of Lemma 2 and the proof sketch of Theorem 1 should be kept as is in the main text, as opposed to moving them to the Appendix.
>
> Now, we will provide explicit answers to the questions in the review:
>
> Question 1: While in the paper the approach focuses on the eigenvectors of the graph Laplacian, in the experiments it is used for finding eigenfunctions. I think that further information should be provided for the actual formulation/solution of this problem.
>
> Answer: The term eigenfunction is equivalent to the term eigenvector of the graph Laplacian in the abstract setting, i.e., when the state space is not necessarily a finite set. In this sense, our proposed objective can be used to approximate the eigenfunctions of the Laplacian, or what we refer to as the Laplacian representation. We added a new last paragraph in the Background section that we hope clarifies this point. In addition, we extended our original proofs to the abstract setting in the Appendix.
>
> Question 2: I find Corollary 1 and the paragraph above a bit unclear. Why does an optimum of (2) and (4) imply that the constraint must be violated?
>
> Answer: Consider Equation 7, it states that the gradient consists of 3 terms: a Laplacian product $Lu_i$, a sum of dual terms, and a sum of barrier terms. The proof of Lemma 2 relies on the fact that the dual sum goes from the index $j=1$ to $j=i$, and that the barrier loss sum is 0 in the equilibrium points. In cases where the dual variables are not used, that is, when $\beta_{ij}=0$, and stop-gradient operators are not used (i.e., in both GDO and GGDO), the barrier sum cannot be 0 since it needs to cancel the product term $Lu_i$. Hence, if we assume that $u_i$ is an eigenvector of $L$, we have that $u_i$ must lie in the span of the first d eigenvectors $u_1,\cdots,u_d$. Thus, we can conclude that the inner product between $u_i$ and some other eigenvector $u_j$ must be different from the Kronecker delta, as stated in the Corollary 1. Now you can find a proof in Appendix A.3.
>
> Question 3: Perhaps, an experiment to test the stability of the equilibrium with respect to permutations.
>
> Answer: We added Figure 8 in Appendix D that shows how after permuting the coordinates of the Laplacian representation, optimizing ALLO results in recovering the correct ordering. This shows that no permutation is stable besides the desired one.
>
> Question 4: Why rotated eigenvectors do not provide a good representation?
>
> Answer: There are two particular reasons that motivate learning the eigenvectors exactly and not its rotations. First, consider their use as intrinsic rewards to induce option discovery. In this case, the idea is to find a policy that maximizes the cumulative reward determined by each eigenvector, with the characteristic that, once a local maxima is reached, the agent stops using the option. This means that the number of local maxima in the eigenvectors (and minima, since the negative of each eigenvector is also an eigenvector) affects the number of sink states where the options are stopped. When the eigenvectors are rotated, the number of local maxima increases, in general, and so does the number of sink states. The result is that the trajectories traversed by the agent are shorter and so, effectively, the less exploratory that the agent behaves (for a similar intuition, see Jinnai, 2019). In the same vein, rotations correspond to combinations of eigenvectors, which results in different temporal scales being combined. The eigenvalues can be used to normalize these scales for reward shaping, but once there are multiple scales, no normalization is possible. Thus, rotations are not as effective for this use case. To provide some evidence of this phenomena and make things more intuitive, we now include Figure 7 in Appendix C. Moreover, to make it more explicit, now we also briefly mention the problem with using rotations in the fifth paragraph of Section 1.

---

> > ### Comment · Reviewer_nARE · 2023-11-22
> > **Post-rebuttal**
> >
> > I would like to thank the authors for the answers and for updating the manuscript accordingly. Regarding Q1 probably my formulation was not clear enough, but the question was about the actual implementation differences, while the clarification has been already included in the updated paper. After reading the rest of the reviews and the associated answers, I acknowledge that the paper is a good contribution and within the standards of the conference, so I will keep my score 6. I recommend however the authors to take into account the reviews and further update the paper, for example including a few additional experimental results and potentially further clarifications where necessary.

---

### Meta-Review · Area_Chair_DypR · 2023-12-09

**Metareview:**

The paper introduces a new method for approximating the graph Laplacian's eigenvalues and eigenvectors without reliance on sensitive hyperparameters. It presents a reformulated minimax objective that theoretically ensures the unique stable equilibrium point is the Laplacian representation (namely eigenvectors), which is important for learning representations in reinforcement learning. The paper claims this objective is both hyperparameter-independent and theoretically sound. Some experiments are provided to justify the new method.

Strengths:
1) Theoretical side: The proposed method provides a new interesting approach and the paper provides a strong theoretical underpinning for its proposed objective, showing that it converges to the correct solutions independently of hyperparameters.

2) Practical Implementation: The method is demonstrated to work well in experiments and is shown to be applicable to high-dimensional problems, which is valuable for reinforcement learning applications.

Weakness:

1) Clarity in Application: Reviewers initially found it unclear how the theoretical results related to deep learning implementation, suggesting a potential gap in the presentation.

2) Demonstration on Simple Environments: There was some criticism that the environments used in the initial experiments were too simple, though this was later addressed by including more complex, pixel-based environments in subsequent experiments.

3) Stability Analysis: One reviewer pointed out that the stability analysis of the proposed objective might be insufficient, although the authors believe Theorem 1 addresses this concern.

Three reviewers appreciated the novel reformulation, despite of those concerns on the current manuscript. They gave scores above the acceptance borderline. One reviewer submitted an irrelevant or wrong review, and did not response in and after the discussion period, which is thus ignored.

**Justification For Why Not Higher Score:**

Except for the wrong or irrelevant review, three reviewers unanimously agree that the paper is above the acceptance borderline.

**Justification For Why Not Lower Score:**

Three reviewers unanimously agree that the paper is above the acceptance borderline. One reviewer submitted an irrelevant or wrong review, and did not response in and after the discussion period, which is thus ignored.

---

### Decision · Program_Chairs · 2024-01-16

Accept (poster)